# Efficient Gradient Flows in Sliced-Wasserstein Space

**Clément Bonet**                                       *clement.bonet@univ-ubs.fr*
*Université Bretagne Sud, CNRS, LMBA, Vannes, France*

**Nicolas Courty**                                      *nicolas.courty@irisa.fr*
*Université Bretagne Sud, CNRS, IRISA, Vannes, France*

**François Septier**                                    *francois.septier@univ-ubs.fr*
*Université Bretagne Sud, CNRS, LMBA, Vannes, France*

**Lucas Drumetz**                                       *lucas.drumetz@imt-atlantique.fr*
*IMT Atlantique, CNRS, Lab-STICC, Brest, France*

**Reviewed on OpenReview:** *https://openreview.net/forum?id=Au1LNKmRvh*

## Abstract

Minimizing functionals in the space of probability distributions can be done with Wasserstein gradient flows. To solve them numerically, a possible approach is to rely on the Jordan–Kinderlehrer–Otto (JKO) scheme which is analogous to the proximal scheme in Euclidean spaces. However, it requires solving a nested optimization problem at each iteration, and is known for its computational challenges, especially in high dimension. To alleviate it, very recent works propose to approximate the JKO scheme leveraging Brenier's theorem, and using gradients of Input Convex Neural Networks to parameterize the density (JKO-ICNN). However, this method comes with a high computational cost and stability issues. Instead, this work proposes to use gradient flows in the space of probability measures endowed with the sliced-Wasserstein (SW) distance. We argue that this method is more flexible than JKO-ICNN, since SW enjoys a closed-form differentiable approximation. Thus, the density at each step can be parameterized by any generative model which alleviates the computational burden and makes it tractable in higher dimensions.

## 1 Introduction

Minimizing functionals with respect to probability measures is a ubiquitous problem in machine learning. Important examples are generative models such as GANs (Goodfellow et al., 2014; Arjovsky et al., 2017; Lin et al., 2021), VAEs (Kingma & Welling, 2013) or normalizing flows (Papamakarios et al., 2019).

To that aim, one can rely on Wasserstein gradient flows (WGF) (Ambrosio et al., 2008) which are curves decreasing the functional as fast as possible (Santambrogio, 2017). For particular functionals, these curves are known to be characterized by the solution of some partial differential equation (PDE) (Jordan et al., 1998). Hence, to solve Wasserstein gradient flows numerically, we can solve the related PDE when it is available. However, solving a PDE can be a difficult and computationally costly task, especially in high dimension (Han et al., 2018). Fortunately, several alternatives exist in the literature. For example, one can approximate instead a counterpart stochastic differential equation (SDE) related to the PDE followed by the gradient flow. For the Kullback-Leibler divergence, it comes back to the so called unadjusted Langevin algorithm (ULA) (Roberts & Tweedie, 1996; Wibisono, 2018), but it has also been proposed for other functionals such as the Sliced-Wasserstein distance with an entropic regularization (Liutkus et al., 2019).

Another way to solve Wasserstein gradient flows numerically is to approximate the curve in discrete time. By using the well-known forward Euler scheme, particle schemes have been derived for diverse functionals such as the Kullback-Leibler divergence (Feng et al., 2021; Wang et al., 2021; 2022), the maximum mean discrepancy

(Arbel et al., 2019), the kernel Stein discrepancy (Korba et al., 2021) or KALE (Glaser et al., 2021). Salim et al. (2020) propose instead a forward-backward discretization scheme analogously to the proximal gradient algorithm (Bauschke et al., 2011). Yet, these methods only provide samples approximately following the gradient flow, but without any information about the underlying density.

Another time discretization possible is the so-called JKO scheme introduced in (Jordan et al., 1998), which is analogous in probability space to the well-known proximal operator (Parikh & Boyd, 2014) in Hilbertian space and which corresponds to the backward Euler scheme. However, as a nested minimization problem, it is a difficult problem to handle numerically. Some works use a discretization in space (*e.g.* a grid) and the entropic regularization of the Wasserstein distance (Peyré, 2015; Carlier et al., 2017), which benefits from specific resolution strategies. However, those approaches do not scale to high dimensions, as the discretization of the space scales exponentially with the dimension. Very recently, it was proposed in several concomitant works (Alvarez-Melis et al., 2021; Mokrov et al., 2021; Bunne et al., 2022) to take advantage of Brenier's theorem (Brenier, 1991) and model the optimal transport map (Monge map) as the gradient of a convex function with Input Convex Neural Networks (ICNN) (Amos et al., 2017). By solving the JKO scheme with this parametrization, these models, called JKO-ICNN, handle higher dimension problems well. Yet, a drawback of JKO-ICNN is the training time due to a number of evaluations of the gradient of each ICNN that is quadratic in the number of JKO iterations. It also requires to backpropagate through the gradient which is challenging in high dimension, even though stochastic methods were proposed in (Huang et al., 2020) to alleviate it. Moreover, it has also been observed in several works that ICNNs have a poor expressiveness (Rout et al., 2021; Korotin et al., 2019; 2021) and that we should rather directly estimate the gradient of convex functions by neural networks (Saremi, 2019; Richter-Powell et al., 2021). Other recent works proposed to use the JKO scheme by either exploiting variational formulations of functionals in order to avoid the evaluation of densities and allowing to use more general neural networks in (Fan et al., 2021), or by learning directly the density in (Hwang et al., 2021).

In parallel, it was proposed to endow the space of probability measures with other distances than Wasserstein. For example, Gallouët & Monsaingeon (2017) study a JKO scheme in the space endowed by the Kantorovitch-Fisher-Rao distance. However, this still requires a costly JKO step. Several particles schemes were derived as gradient flows into this space (Lu et al., 2019; Zhang et al., 2021). We can also cite Kalman-Wasserstein gradient flows (Garbuno-Inigo et al., 2020) or the Stein variational gradient descent (Liu & Wang, 2016; Liu, 2017; Duncan et al., 2019) which can be seen as gradient flows in the space of probabilities endowed by a generalization of the Wasserstein distance. However, the JKO schemes of these different metrics are not easily tractable in practice.

**Contributions.** In the following, we propose to study the JKO scheme in the space of probability distributions endowed with the sliced-Wasserstein (SW) distance (Rabin et al., 2011). This novel and simple modification of the original problem comes with several benefits, mostly linked to the fact that this distance is easily differentiable and computationally more tractable than the Wasserstein distance. We first derive some properties of this new class of flows and discuss links with Wasserstein gradient flows. Notably, we observe empirically for both gradient flows the same dynamic, up to a time dilation of parameter the dimension of the space. Then, we show that it is possible to minimize functionals and learn the stationary distributions in high dimensions, on toy datasets as well as real image datasets, using *e.g.* neural networks. In particular, we propose to use normalizing flows for functionals which involve the density, such as the negative entropy. Finally, we exhibit several examples for which our strategy performs better than JKO-ICNN, either *w.r.t.* to computation times and/or *w.r.t.* the quality of the final solutions.

## 2  Background

In this paper, we are interested in finding a numerical solution to gradient flows in probability spaces. Such problems generally arise when minimizing a functional $\mathcal{F}$ defined on $\mathcal{P}(\mathbb{R}^d)$, the set of probability measures on $\mathbb{R}^d$:

$$\min_{\mu \in \mathcal{P}(\mathbb{R}^d)} \mathcal{F}(\mu), \tag{1}$$

but they can also be defined implicitly through their dynamics, expressed as partial differential equations. JKO schemes are implicit optimization methods that operate on particular discretizations of these problems and consider the natural metric of $\mathcal{P}(\mathbb{R}^d)$ to be Wasserstein. Recalling our goal is to study similar schemes with an alternative, computationally friendly metric (SW), we start by defining formally the notion of gradient flows in Euclidean spaces, before switching to probability spaces. We finally give a rapid overview of existing numerical schemes.

## 2.1 Gradient Flows in Euclidean Spaces

Let $F : \mathbb{R}^d \to \mathbb{R}$ be a functional. A gradient flow of $F$ is a curve (*i.e.* a continuous function from $\mathbb{R}_+$ to $\mathbb{R}^d$) which decreases $F$ as much as possible along it. If $F$ is differentiable, then a gradient flow $x : [0, T] \to \mathbb{R}^d$ solves the following Cauchy problem (Santambrogio, 2017)

$$\begin{cases} \frac{\mathrm{d}x(t)}{\mathrm{d}t} = -\nabla F(x(t)), \\ x(0) = x_0. \end{cases} \tag{2}$$

Under conditions on $F$ (*e.g.* $\nabla F$ Lipschitz continuous, $F$ convex or semi-convex), this problem admits a unique solution which can be approximated using numerical schemes for ordinary differential equations such as the explicit or the implicit Euler scheme. For the former, we recover the regular gradient descent, and for the latter, we recover the proximal point algorithm (Parikh & Boyd, 2014): let $\tau > 0$,

$$x_{k+1}^\tau \in \arg\min_x \ \frac{\|x - x_k^\tau\|_2^2}{2\tau} + F(x) = \mathrm{prox}_{\tau F}(x_k^\tau). \tag{3}$$

This formulation does not use any gradient, and can therefore be used in any metric space by replacing $\|\cdot\|_2$ with the right distance.

## 2.2 Gradient Flows in Probability Spaces

To define gradient flows in the space of probability measures, we first need a metric. We restrict our analysis to probability measures with moments of order 2: $\mathcal{P}_2(\mathbb{R}^d) = \{\mu \in \mathcal{P}(\mathbb{R}^d), \ \int \|x\|^2 \mathrm{d}\mu(x) < +\infty\}$. Then, a possible distance on $\mathcal{P}_2(\mathbb{R}^d)$ is the Wasserstein distance (Villani, 2008), let $\mu, \nu \in \mathcal{P}_2(\mathbb{R}^d)$,

$$W_2^2(\mu, \nu) = \min_{\gamma \in \Pi(\mu,\nu)} \ \int \|x - y\|_2^2 \ \mathrm{d}\gamma(x, y), \tag{4}$$

where $\Pi(\mu, \nu)$ is the set of probability measures on $\mathbb{R}^d \times \mathbb{R}^d$ with marginals $\mu$ and $\nu$.

Now, by endowing the space of measures with $W_2$, we can define the Wasserstein gradient flow of a functional $\mathcal{F} : \mathcal{P}_2(\mathbb{R}^d) \to \mathbb{R}$ by plugging $W_2$ in (3) which becomes

$$\mu_{k+1}^\tau \in \arg\min_{\mu \in \mathcal{P}_2(\mathbb{R}^d)} \ \frac{W_2^2(\mu, \mu_k^\tau)}{2\tau} + \mathcal{F}(\mu). \tag{5}$$

The gradient flow is then the limit of the sequence of minimizers when $\tau \to 0$. This scheme was introduced in the seminal work of Jordan et al. (1998) and is therefore referred to as the JKO scheme. In this work, the authors showed that gradient flows are linked to PDEs, and in particular with the Fokker-Planck equation when the functional $\mathcal{F}$ is of the form

$$\mathcal{F}(\mu) = \int V \mathrm{d}\mu + \mathcal{H}(\mu) \tag{6}$$

where $V$ is some potential function and $\mathcal{H}$ is the negative entropy: let $\sigma$ denote the Lebesgue measure,

$$\mathcal{H}(\mu) = \begin{cases} \int \rho(x) \log(\rho(x)) \ \mathrm{d}x & \text{if } \mathrm{d}\mu = \rho \mathrm{d}\sigma \\ +\infty & \text{otherwise.} \end{cases} \tag{7}$$

Then, the limit of $(\mu^\tau)_\tau$ when $\tau \to 0$ is a curve $t \mapsto \mu_t$ such that for all $t > 0$, $\mu_t$ has a density $\rho_t$. The curve $\rho$ satisfies (weakly) the Fokker-Planck PDE

$$\frac{\partial \rho}{\partial t} = \text{div}(\rho \nabla V) + \Delta \rho. \tag{8}$$

For more details on gradient flows in metric space and in Wasserstein space, we refer to (Ambrosio et al., 2008). Note that many other functional can be plugged in equation 5 defining different PDEs. We introduce here the Fokker-Planck PDE as a classical example, since the functional is connected to the Kullback-Leibler (KL) divergence and its Wasserstein gradient flow is connected to many classical algorithms such as the unadjusted Langevin algorithm (ULA) (Wibisono, 2018). But we will also use other functionals in Section 4 such as the interaction functional defined for regular enough $W$ as

$$\mathcal{W}(\mu) = \frac{1}{2} \iint W(x - y) \; d\mu(x) d\mu(y), \tag{9}$$

which admits as Wasserstein gradient flow the aggregation equation (Santambrogio, 2015, Chapter 8)

$$\frac{\partial \rho}{\partial t} = \text{div}\big(\rho(\nabla W * \rho)\big) \tag{10}$$

where $*$ denotes the convolution operation.

### 2.3   Numerical Methods to solve the JKO Scheme

Solving (5) is not an easy problem as it requires to solve an optimal transport problem at each step and hence is composed of two nested minimization problems.

There are several strategies which were used to tackle this problem. For example, Laborde (2016) rewrites (5) as a convex minimization problem using the Benamou-Brenier dynamic formulation of the Wasserstein distance (Benamou & Brenier, 2000). Peyré (2015) approximates the JKO scheme by using the entropic regularization and rewriting the problem with respect to the Kullback-Leibler proximal operator. The problem becomes easier to solve using Dykstra's algorithm (Dykstra, 1985). This scheme was proved to converge to the right PDE in (Carlier et al., 2017). It was proposed to use the dual formulation in other works such as (Caluya & Halder, 2019) or (Frogner & Poggio, 2020). Cancès et al. (2020) proposed to linearize the Wasserstein distance using the weighted Sobolev approximation (Villani, 2003; Peyre, 2018).

More recently, following (Benamou et al., 2016), Alvarez-Melis et al. (2021) and Mokrov et al. (2021) proposed to exploit Brenier's theorem by rewriting the JKO scheme as

$$u_{k+1}^\tau \in \underset{u \text{ convex}}{\arg\min} \; \frac{1}{2\tau} \int \|\nabla u(x) - x\|_2^2 \; d\mu_k^\tau(x) + \mathcal{F}\big((\nabla u)_{\#}\mu_k^\tau\big) \tag{11}$$

and model the probability measures as $\mu_{k+1}^\tau = (\nabla u_{k+1}^\tau)_{\#}\mu_k^\tau$ where $\#$ is the push forward operator, defined as $(\nabla u)_{\#}\mu_k^\tau(A) = \mu_k^\tau((\nabla u)^{-1}(A))$ for all $A \in \mathcal{B}(\mathbb{R}^d)$. Then, to solve it numerically, they model convex functions using ICNNs (Amos et al., 2017):

$$\theta_{k+1}^\tau \in \underset{\theta \in \{\theta, u_\theta \in \text{ICNN}\}}{\arg\min} \; \frac{1}{2\tau} \int \|\nabla_x u_\theta(x) - x\|_2^2 \; d\mu_k^\tau(x) + \mathcal{F}\big((\nabla_x u_\theta)_{\#}\mu_k^\tau\big). \tag{12}$$

In the remainder, this method is denoted as JKO-ICNN. Bunne et al. (2022) also proposed to use ICNNs into the JKO scheme, but with a different objective of learning the functional from samples trajectories along the timesteps. Lastly, Fan et al. (2021) proposed to learn directly the Monge map $T$ by solving at each step the following problem:

$$T_{k+1}^\tau \in \underset{T}{\arg\min} \; \frac{1}{2\tau} \int \|T(x) - x\|_2^2 \; d\mu_k^\tau(x) + \mathcal{F}(T_{\#}\mu_k^\tau) \tag{13}$$

and using variational formulations of functionals involving the density. This formulation requires only to use samples from the measure. However, it needs to be derived for each functional, and involves minimax optimization problems which are notoriously hard to train (Arjovsky & Bottou, 2017; Bond-Taylor et al., 2021).

## 3 Sliced-Wasserstein Gradient Flows

As seen in the previous section, solving numerically (5) is a challenging problem. To tackle high-dimensional settings, one could benefit from neural networks, such as generative models, that are known to model high-dimensional distributions accurately. The problem being not directly differentiable, previous works relied on Brenier's theorem and modeled convex functions through ICNNs, which results in JKO-ICNN. However, this method is very costly to train. For a JKO scheme of $k$ steps, it requires $O(k^2)$ evaluations of gradients (Mokrov et al., 2021) which can be a huge price to pay when the dynamic is very long. Moreover, it requires to backpropagate through gradients, and to compute the determinant of the Jacobian when we need to evaluate the likelihood (assuming the ICNN is strictly convex). The method of Fan et al. (2021) also requires $O(k^2)$ evaluations of neural networks, as well as to solve a minimax optimization problem at each step.

Here, we propose instead to use the space of probability measures endowed with the sliced-Wasserstein (SW) distance by modifying adequately the JKO scheme. Surprisingly enough, this class of gradient flows, which are very easy to compute, has never been considered numerically in the literature. Close to our work, **Wasserstein** gradient flows using SW as a functional (called Sliced-Wasserstein flows) have been considered in (Liutkus et al., 2019). Our method differs significantly from this work, since we propose to compute **sliced-Wasserstein** gradient flows of different functionals.

We first introduce SW along with motivations to use this distance. We then study some properties of the scheme and discuss links with Wasserstein gradient flows. Since this metric is known in closed-form, the JKO scheme is more tractable numerically and can be approximated in several ways that we describe in Section 3.3.

### 3.1 Motivations

**Sliced-Wasserstein Distance.**  The Wasserstein distance (4) is generally intractable. However, in one dimension, for $\mu, \nu \in \mathcal{P}_2(\mathbb{R})$, we have the following closed-form (Peyré et al., 2019, Remark 2.30)

$$W_2^2(\mu, \nu) = \int_0^1 \left( F_\mu^{-1}(u) - F_\nu^{-1}(u) \right)^2 \, \mathrm{d}u \tag{14}$$

where $F_\mu^{-1}$ (resp. $F_\nu^{-1}$) is the quantile function of $\mu$ (resp. $\nu$). It motivated the construction of the sliced-Wasserstein distance (Rabin et al., 2011; Bonnotte, 2013), as for $\mu, \nu \in \mathcal{P}_2(\mathbb{R}^d)$,

$$SW_2^2(\mu, \nu) = \int_{S^{d-1}} W_2^2(P_\#^\theta \mu, P_\#^\theta \nu) \, \mathrm{d}\lambda(\theta) \tag{15}$$

where $P^\theta(x) = \langle x, \theta \rangle$ and $\lambda$ is the uniform distribution on $S^{d-1} = \{\theta \in \mathbb{R}^d, \ \|\theta\|_2 = 1\}$.

**Computational Properties.**  Firstly, $SW_2$ is very easy to compute by a Monte-Carlo approximation (see Appendix C.1). It is also differentiable, and hence using *e.g.* the Python Optimal Transport (POT) library (Flamary et al., 2021), we can backpropagate *w.r.t.* parameters or weights parametrizing the distributions (see Section 3.3). Note that some libraries allow to directly backpropagate through Wasserstein. However, theoretically, we only have access to a subgradient in that case (Cuturi & Doucet, 2014, Proposition 1), and the computational complexity is bigger ($O(n^3 \log n)$ versus $O(n \log n)$ for SW). Moreover, contrary to $W_2$, its sample complexity does not depend on the dimension (Nadjahi et al., 2020) which is important to overcome the curse of dimensionality. However, it is known to be hard to approximate in high-dimension (Deshpande et al., 2019) since the error of the Monte-Carlo estimates is impacted by the number of projections in practice (Nadjahi et al., 2020). Nevertheless, there exist several variants which could also be used. Moreover, a deterministic approach using a concentration of measure phenomenon (and hence being more accurate in high dimension) was recently proposed by Nadjahi et al. (2021) to approximate $SW_2$.

**Link with Wasserstein.**  The sliced-Wasserstein distance has also many properties related to the Wasserstein distance. First, they actually induce the same topology (Nadjahi et al., 2019; Bayraktar & Guoï, 2021) which might justify to use SW as a proxy of Wasserstein. Moreover, as showed in Chapter 5 of Bonnotte

(2013), they can be related on compact sets by the following inequalities, let $R > 0$, for all $\mu, \nu \in \mathcal{P}(B(0, R))$,

$$SW_2^2(\mu, \nu) \leq c_d^2 W_2^2(\mu, \nu) \leq C_d^2 SW_2^{\frac{1}{d+1}}(\mu, \nu), \tag{16}$$

with $c_d^2 = \frac{1}{d}$ and $C_d$ some constant.

Hence, from these properties, we can wonder whether their gradient flows are related or not, or even better, whether they are the same or not. This property was initially conjectured by Filippo Santambrogio[1]. Some previous works started to gather some hints on this question. For example, Candau-Tilh (2020) showed that, while $(\mathcal{P}_2(\mathbb{R}^d), SW_2)$ is not a geodesic space, the minimal length (in metric space, Definition 2.4 in (Santambrogio, 2017)) connecting two measures is $W_2$ up to a constant (which is actually $c_d$). We report in Appendix A some results introduced by Candau-Tilh (2020).

## 3.2 Definition and Properties of Sliced-Wasserstein Gradient Flows

Instead of solving the regular JKO scheme (5), we propose to introduce a SW-JKO scheme, let $\mu_0 \in \mathcal{P}_2(\mathbb{R}^d)$,

$$\forall k \geq 0, \ \mu_{k+1}^\tau \in \operatorname*{arg\,min}_{\mu \in \mathcal{P}_2(\mathbb{R}^d)} \frac{SW_2^2(\mu, \mu_k^\tau)}{2\tau} + \mathcal{F}(\mu) \tag{17}$$

in which we replaced the Wasserstein distance by $SW_2$.

To study gradient flows and show that they are well defined, we first have to check that discrete solutions of the problem (17) indeed exist. Then, we have to check that we can pass to the limit $\tau \to 0$ and that the limit satisfies gradient flows properties. These limit curves will be called Sliced-Wasserstein gradient flows (SWGFs).

In the following, we restrain ourselves to measures on $\mathcal{P}_2(K)$ where $K \subset \mathbb{R}^d$ is a compact set. We report some properties of the scheme (17) such as existence and uniqueness of the minimizer, and refer to Appendix B for the proofs.

**Proposition 1.** *Let $\mathcal{F} : \mathcal{P}_2(K) \to \mathbb{R}$ be a lower semi continuous functional, then the scheme (17) admits a minimizer. Moreover, it is unique if $\mu_k^\tau$ is absolutely continuous and $\mathcal{F}$ convex or if $\mathcal{F}$ is strictly convex.*

This proposition shows that the problem is well defined for convex lower semi continuous functionals since we can find at least a minimizer at each step. The assumptions on $\mathcal{F}$ are fairly standard and will apply for diverse functionals such as for example (6) or (9) for $V$ and $W$ regular enough.

**Proposition 2.** *The functional $\mathcal{F}$ is non increasing along the sequence of minimizers $(\mu_k^\tau)_k$.*

As the ultimate goal is to find the minimizer of the functional, this proposition assures us that the solution will decrease $\mathcal{F}$ along it at each step. If $\mathcal{F}$ is bounded below, then the sequence $\left(\mathcal{F}(\mu_k^\tau)\right)_k$ will converge (since it is non increasing).

More generally, by defining the piecewise constant interpolation as $\mu^\tau(0) = \mu_0$ and for all $k \geq 0$, $t \in \,]k\tau, (k+1)\tau]$, $\mu^\tau(t) = \mu_{k+1}^\tau$, we can show that for all $t < s$, $SW_2(\mu^\tau(t), \mu^\tau(s)) \leq C\left(|t-s|^{\frac{1}{2}} + \tau^{\frac{1}{2}}\right)$. Following Santambrogio (2017), we can apply the Ascoli-Arzelà theorem (Santambrogio, 2015, Box 1.7) and extract a converging subsequence. However, the limit when $\tau \to 0$ is possibly not unique and has a priori no relation with $\mathcal{F}$. Since $(\mathcal{P}_2(\mathbb{R}^d), SW_2)$ is not a geodesic space, but rather a "pseudo-geodesic" space whose true geodesics are $c_d W_2$ (Candau-Tilh, 2020) (see Appendix A.1), we cannot directly apply the theory introduced in (Ambrosio et al., 2008). We leave for future works the study of the theoretical properties of the limit. Nevertheless, we conjecture that in the limit $t \to \infty$, SWGFs converge toward the same measure as for WGFs. We will study it empirically in Section 4 by showing that we are able to find as good minima as WGFs for different functionals.

**Limit PDE.** We discuss here some possible links between SWGFs and WGFs. Candau-Tilh (2020) shows that the Euler-Lagrange equation of the functional (6) has a similar form (up to the first variation of the

---

[1]in private communications

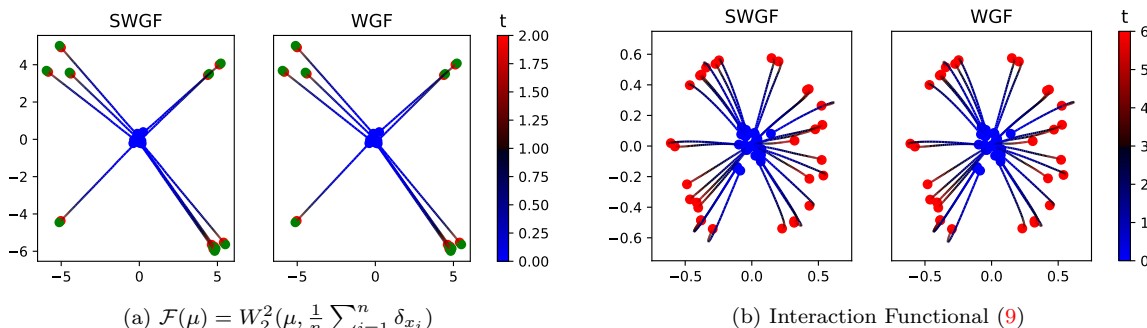

(a) $\mathcal{F}(\mu) = W_2^2(\mu, \frac{1}{n}\sum_{i=1}^n \delta_{x_i})$        (b) Interaction Functional (9)

Figure 1: Comparison of the trajectories of (dilated by $d = 2$) Sliced-Wasserstein gradient flows (SWGF) and Wasserstein gradient flows (WGF) of different functionals. On the left, the stationary solution is a uniform discrete distributions. On the right, the stationary solution is a Dirac ring of radius 0.5. Blue points represent the initial positions, red points the final positions, and green points the target particles.

distance, see Appendix A.2). Hence, he conjectures that there is a correlation between the two gradient flows. We identify here some cases for which we can relate the Sliced-Wasserstein gradient flows to the Wasserstein gradient flows.

We first notice that for one dimensional supported measures, $W_2$ and $SW_2$ are the same up to a constant $\sqrt{d}$, $i.e.$ let $\mu, \nu \in \mathcal{P}_2(\mathbb{R}^d)$ be supported on the same line, then $SW_2^2(\mu, \nu) = W_2^2(\mu, \nu)/d$. Interestingly enough, this is the same constant as between geodesics. This property is actually still true in any dimension for Gaussians with a covariance matrix of the form $cI_d$ with $c > 0$. Therefore, we argue that for these classes of measures, provided that the minimum at each step stays in the same class, we would have a dilation of factor $d$ between the WGF and the SWGF. For example, for the Fokker-Planck functional, the PDE followed by the SWGF would become $\frac{\partial \rho}{\partial t} = d(\text{div}(\rho\nabla V) + \Delta\rho)$. And, by correcting the SW-JKO scheme as

$$\mu_{k+1}^\tau \in \underset{\mu \in \mathcal{P}_2(\mathbb{R}^d)}{\arg\min} \ \frac{d}{2\tau} SW_2^2(\mu, \mu_k^\tau) + \mathcal{F}(\mu), \tag{18}$$

we would have the same dynamic. For more general measures, it is not the case anymore. But, by rewriting $SW_2^2$ and $W_2^2$ $w.r.t.$ the centered measures $\bar{\mu}$ and $\bar{\nu}$, as well as the means $m_\mu = \int x \, d\mu(x)$ and $m_\nu = \int x \, d\nu(x)$, we have:

$$W_2^2(\mu, \nu) = \|m_\mu - m_\nu\|_2^2 + W_2^2(\bar{\mu}, \bar{\nu}), \quad SW_2^2(\mu, \nu) = \frac{\|m_\mu - m_\nu\|_2^2}{d} + SW_2^2(\bar{\mu}, \bar{\nu}). \tag{19}$$

Hence, for measures characterized by their mean and variance ($e.g.$ Gaussians), there will be a constant $d$ between the optimal mean of the SWGF and of the WGF. However, such a direct relation is not available between variances, even on simple cases like Gaussians. We report in Appendix B.4 the details of the calculations.

We draw on Figure 1 trajectories of SWGFs and WGFs with $\mathcal{F}(\mu) = W_2^2(\mu, \frac{1}{n}\sum_{i=1}^n \delta_{x_i})$ and $\mathcal{F}$ as in (9) with $W$ chosen as in Section 4.2. For the former, the target is a discrete measure with uniform weights and, using the same number of particles in the approximation $\hat{\mu}_n$, and performing gradient descent on the particles as explained in Section 3.3, we expect the Wasserstein gradient flow to push each particle on the closest target particle. This is indeed what we observe. For the latter, the stationary solution is a Dirac ring, as further explained in Section 4.2. In both cases, by using a dilation parameter of $d$, we observe almost the same trajectories between SWGF and WGF, which is an additional support of the conjecture that the trajectories of the gradient flows in both spaces are alike. We also report in Appendix D.1 evolutions along the approximated WGF and SWGF of different functionals.

## 3.3 Solving the SW-JKO Scheme in Practice

As a Monte-Carlo approximate of SW can be computed in closed-form, equation 17 is not a nested minimization problem anymore and is differentiable. We present here a few possible parameterizations of probability

---

**Algorithm 1** SW-JKO with Generative Models

---

**Input:** $\mu_0$ the initial distribution, $K$ the number of SW-JKO steps, $\tau$ the step size, $\mathcal{F}$ the functional, $N_e$ the number of epochs to solve each SW-JKO step, $N$ the batch size

**for** $k = 1$ **to** $K$ **do**

    Initialize a neural network $g_\theta^{k+1}$ *e.g.* with $g_\theta^k$

    **for** $i = 1$ **to** $N_e$ **do**

        Sample $z_j^{(k)}, z_j^{(k+1)} \sim p_Z$ i.i.d

        $x_j^{(k)} = g_\theta^k(z_j^{(k)})$, $x_j^{(k+1)} = g_\theta^{k+1}(z_j^{(k+1)})$

        // Denote $\hat{\mu}_k^\tau = \frac{1}{N} \sum_{j=1}^N \delta_{x_j^{(k)}}$, $\hat{\mu}_{k+1}^\tau = \frac{1}{N} \sum_{j=1}^N \delta_{x_j^{(k+1)}}$

        $J(\hat{\mu}_{k+1}^\tau) = \frac{1}{2\tau} SW_2^2(\hat{\mu}_k^\tau, \hat{\mu}_{k+1}^\tau) + \mathcal{F}(\hat{\mu}_{k+1}^\tau)$

        Backpropagate through $J$ *w.r.t.* $\theta$

        Perform a gradient step using *e.g.* Adam

    **end for**

**end for**

---

distributions which we can use in practice through SW-JKO to approximate the gradient flow. We further state, as an example, how to approximate the Fokker-Planck functional (6). Indeed, classical other functionals can be approximated using the same method since they often only require to approximate an integral *w.r.t.* the measure of interests and to evaluate its density as for (6). Then, from these parameterizations, we can apply gradient-based optimization algorithms by using backpropagation over the loss at each step.

**Discretized Grid.** A first proposition is to model the distribution on a regular fixed grid, as it is done *e.g.* in (Peyré, 2015). If we approximate the distribution by a discrete distribution with a fixed grid on which the different samples are located, then we only have to learn the weights. Let us denote $\mu_k^\tau = \sum_{i=1}^N \rho_i^{(k)} \delta_{x_i}$ where we use $N$ samples located at $(x_i)_{i=1}^N$, and $\sum_{i=1}^N \rho_i = 1$. Let $\Sigma_N$ denote the simplex, then the optimization problem (17) becomes:

$$\min_{(\rho_i)_i \in \Sigma_N} \frac{SW_2^2(\sum_{i=1}^N \rho_i \delta_{x_i}, \mu_k^\tau)}{2\tau} + \mathcal{F}\big(\sum_{i=1}^N \rho_i \delta_{x_i}\big). \tag{20}$$

The entropy is only defined for absolutely continuous distributions. However, following (Carlier et al., 2017; Peyré, 2015), we can approximate the Lebesgue measure as: $L = l \sum_{i=1}^N \delta_{x_i}$ where $l$ represents a volume of each grid point (we assume that each grid point represents a volume element of uniform size). In that case, the Lebesgue density can be approximated by $(\frac{\rho_i}{l})_i$. Hence, for the Fokker-Planck (6) example, we approximate the potential and internal energies as

$$\mathcal{V}(\mu) = \int V(x)\rho(x)\mathrm{d}x \approx \sum_{i=1}^N V(x_i)\rho_i, \quad \mathcal{H}(\mu) = \int \log\big(\rho(x)\big)\rho(x)\mathrm{d}x \approx \sum_{i=1}^N \log\big(\frac{\rho_i}{l}\big)\rho_i. \tag{21}$$

To stay on the simplex, we use a projected gradient descent (Condat, 2016). A drawback of discretizing the grid is that it becomes intractable in high dimension.

**With Particles.** We can also optimize over the position of a set of particles, assigning them uniform weights: $\mu_k^\tau = \frac{1}{N} \sum_{i=1}^N \delta_{x_i^{(k)}}$. The problem (17) becomes:

$$\min_{(x_i)_i} \frac{SW_2^2\big(\frac{1}{N} \sum_{i=1}^N \delta_{x_i}, \mu_k^\tau\big)}{2\tau} + \mathcal{F}\big(\frac{1}{N} \sum_{i=1}^N \delta_{x_i}\big). \tag{22}$$

In that case however, we do not have access to the density and cannot directly approximate $\mathcal{H}$ (or more generally internal energies). A workaround is to use nonparametric estimators (Beirlant et al., 1997), which is however impractical in high dimension.

**Generative Models.** Another solution to model the distribution is to use generative models. Let us denote $g_\theta : \mathcal{Z} \to \mathcal{X}$ such a model, with $\mathcal{Z}$ a latent space, $\theta$ the parameters of the model that will be learned, and let $p_Z$ be a simple distribution (*e.g.* Gaussian). Then, we will denote $\mu^\tau_{k+1} = (g^{k+1}_\theta)_\# p_Z$. The SW-JKO scheme (17) will become in this case

$$\min_\theta \ \frac{SW_2^2\big((g^{k+1}_\theta)_\# p_Z, \mu^\tau_k\big)}{2\tau} + \mathcal{F}\big((g^{k+1}_\theta)_\# p_Z\big). \tag{23}$$

To approximate the negative entropy, we have to be able to evaluate the density. A straightforward choice that we use in our experiments is to use invertible neural networks with a tractable density such as normalizing flows (Papamakarios et al., 2019; Kobyzev et al., 2020). Another solution could be to use the variational formulation as in (Fan et al., 2021) as we only need samples in that case, but at the cost of solving a minimax problem.

To perform the optimization, we can sample points of the different distributions at each step and use a Monte-Carlo approximation in order to approximate the integrals. Let $z_i \sim p_Z$ i.i.d, then $g_\theta(z_i) \sim (g_\theta)_\# p_Z = \mu$ and

$$\mathcal{V}(\mu) \approx \frac{1}{N} \sum_{i=1}^N V(g_\theta(z_i)), \quad \mathcal{H}(\mu) \approx \frac{1}{N} \sum_{i=1}^N \big(\log(p_Z(z_i)) - \log|\det(J_{g_\theta}(z_i))|\big). \tag{24}$$

using the change of variable formula in $\mathcal{H}$.

We sum up the procedure when modeling distributions with generative models in Algorithm 1. We provide the algorithms for the discretized grid and for the particles in Appendix C.2.

**Complexity.** Denoting by $d$ the dimension, $K$ the number of outer iterations, $N_e$ the number of inner optimization step, $N$ the batch size and $L$ the number of projections to approximate SW, SW-JKO has a complexity of $O(KN_eLN\log N)$ versus $O(KN_e((K+d)N + d^3))$ for JKO-ICNN (Mokrov et al., 2021) and $O(K^2N_eNN_m)$ for the variational formulation of Fan et al. (2021) where $N_m$ denotes the number of maximization iteration. Hence, we see that the SW-JKO scheme is more appealing for problems which will require very long dynamics.

**Direct Minimization.** A straightforward way to minimize a functional $\mu \mapsto \mathcal{F}(\mu)$ would be to parameterize the distributions as described in this section and then to perform a direct minimization of the functional by performing a gradient descent on the weights, *i.e.* for instance with a generative model, solving $\min_\theta \ \mathcal{F}((g_\theta)_\# p_z)$. While it is a viable solution, we noted that this is not much discussed in related papers implementing Wasserstein gradient flows with neural networks via the JKO scheme. This problem is theoretically not well defined as a gradient flow on the space of probability measures. And hence, it has less theoretical guarantees of convergence than Wasserstein gradient flows. In our experiments, we noted that the direct minimization suffers from more numerical instabilities in high dimensions, while SW acts as a regularizer. For simpler problems however, the performances can be pretty similar.

## 4 Experiments

In this section, we show that by approximating sliced-Wasserstein gradient flows using the SW-JKO scheme (17), we are able to minimize functionals as well as Wasserstein gradient flows approximated by the JKO-ICNN scheme and with a better computational complexity. We first evaluate the ability to learn the stationary density for the Fokker-Planck equation (8) in the Gaussian case, and in the context of Bayesian Logistic Regression. Then, we evaluate it on an Aggregation equation. Finally, we use SW as a functional with image datasets as target, and compare the results with Sliced-Wasserstein flows introduced in (Liutkus et al., 2019).

For these experiments, we mainly use generative models. When it is required to evaluate the density (*e.g.* to estimate $\mathcal{H}$), we use Real Non Volume Preserving (RealNVP) normalizing flows (Dinh et al., 2016). Our experiments were conducted using PyTorch (Paszke et al., 2019).

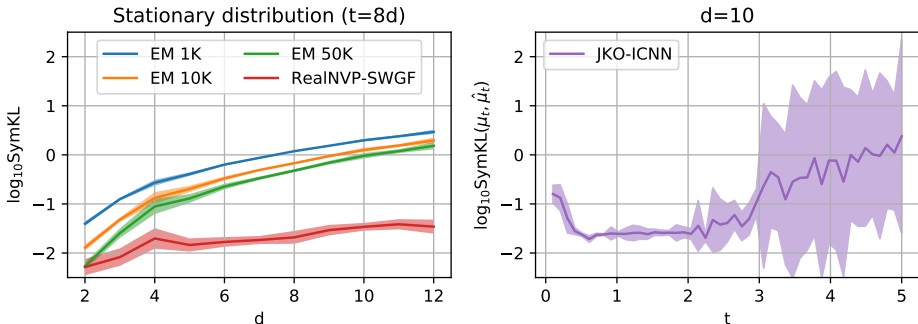

Figure 2: On the left, SymKL divergence between solutions at time $t = 8d$ (using $\tau = 0.1$ and 80 steps in equation 17) and stationary measure. On the right, SymKL between the true WGF $\mu_t$ and the approximation with JKO-ICNN $\hat{\mu}_t$, run through 3 Gaussians with $\tau = 0.1$. We observe instabilities at some point.

### 4.1 Convergence to Stationary Distribution for the Fokker-Planck Equation

We first focus on the functional (6). Its Wasserstein gradient flow is solution of a PDE of the form of (8). In this case, it is well known that the solution converges as $t \to \infty$ towards a unique stationary measure $\mu^* \propto e^{-V}$ (Risken, 1996). Hence, we focus here on learning this target distribution. First, we will choose as target a Gaussian, and then in a second experiment, we will learn a posterior distribution in a bayesian logistic regression setting.

**Gaussian Case.** Taking $V$ of the form $V(x) = \frac{1}{2}(x - m)^T A(x - b)$ for all $x \in \mathbb{R}^d$, with $A$ a symmetric positive definite matrix and $m \in \mathbb{R}^d$, then the stationary distribution is $\mu^* = \mathcal{N}(m, A^{-1})$. We plot in Figure 2 the symmetric Kullback-Leibler (SymKL) divergence over dimensions between approximated distributions and the true stationary distribution. We choose $\tau = 0.1$ and performed 80 SW-JKO steps. We take the mean over 15 random gaussians for dimensions $d \in \{2, \ldots, 12\}$ for randomly generated positive semi-definite matrices $A$ using "make_spd_matrix" from scikit-learn (Pedregosa et al., 2011). Moreover, we use RealNVPs in SW-JKO. We compare the results with the Unadjusted Langevin Algorithm (ULA) (Roberts & Tweedie, 1996), called Euler-Maruyama (EM) since it is the EM approximation of the Langevin equation, which corresponds to the counterpart SDE of the PDE (8). We see that, in dimension higher than 2, the results of the SWGF with RealNVP are better than with this particle scheme obtained with a step size of $10^{-3}$ and with either $10^3$, $10^4$ or $5 \cdot 10^4$ particles. We do not plot the results for JKO-ICNN as we observe many instabilities (right plot in Figure 2). Moreover, we notice a very long training time for JKO-ICNN. We add more details in Appendix D.2. We further note that SW acts here as a regularizer. Indeed, by training normalizing flows with the reverse KL (which is equal to equation 6 up to a constant), we obtain similar results, but with much more instabilities in high dimensions.

**Curse of Dimensionality.** Even though the sliced-Wasserstein distance sample complexity does not suffer from the curse of dimensionality, it appears through the Monte-Carlo approximation (Nadjahi et al., 2020). Here, since SW plays a regularizer role, the objective is not necessarily to approximate it well but rather to minimize the given functional. Nevertheless, the number of projections can still have an impact on the minimization, and we report on Figure 3 the evolution of the found minima *w.r.t.* the number of projections, averaged over 15 random Gaussians. We observe that we do

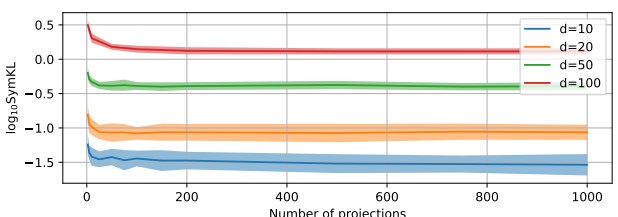

Figure 3: Impact of the number of projections for a fixed number of epochs.

not need much projections to have fairly good results, even in higher dimension. Indeed, with more than 200 projections, the performances stay pretty stables.

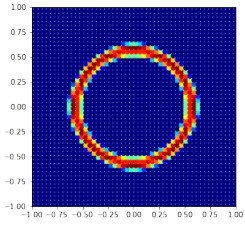
(a) Steady state on the discretized grid

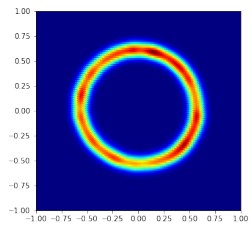
(b) Steady state for the fully connected neural network

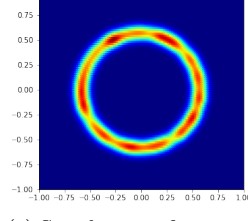
(c) Steady state for particles

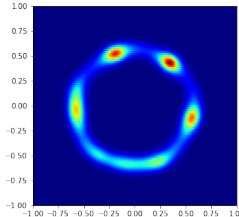
(d) Steady state for JKO-ICNN (with $\tau = 0.1$)

Figure 4: Steady state of the aggregation equation for $a = 4$, $b = 2$. From left to right, we plot it for the discretized grid, for the FCNN, for particles and for JKO-ICNN. We observe that JKO-ICNN does not recover the ring correctly as the particles are not evenly distributed on it.

**Bayesian Logistic Regression.** Following the experiment of Mokrov et al. (2021) in Section 4.3, we propose to tackle the Bayesian Logistic Regression problem using SWGFs. For this task, we want to sample from $p(x|D)$ where $D$ represent data and $x = (w, \log \alpha)$ with $w$ the regression weights on which we apply a Gaussian prior $p_0(w|\alpha) = \mathcal{N}(w; 0, \alpha^{-1})$ and with $p_0(\alpha) = \Gamma(\alpha; 1, 0.01)$. In that case, we use $V(x) = -\log p(x|D)$ to learn $p(x|D)$. We refer to Appendix D.2 for more details on the experiments, as well as hyperparameters. We report in Table 1 the accuracy results obtained on different datasets with SWGFs and compared with JKO-ICNN. We

Table 1: Accuracy and Training Time for Bayesian Logistic Regression over 5 runs

| | JKO-ICNN | | SWGF+RealNVP | |
|---|---|---|---|---|
| Dataset | Acc | t | Acc | t |
| covtype | $0.755 \pm 5 \cdot 10^{-4}$ | 33702s | $0.755 \pm 3 \cdot 10^{-3}$ | 103s |
| german | $0.679 \pm 5 \cdot 10^{-3}$ | 2123s | $\mathbf{0.68} \pm 5 \cdot 10^{-3}$ | 82s |
| diabetis | $0.777 \pm 7 \cdot 10^{-3}$ | 4913s | $\mathbf{0.778} \pm 2 \cdot 10^{-3}$ | 122s |
| twonorm | $0.981 \pm 2 \cdot 10^{-4}$ | 6551s | $0.981 \pm 6 \cdot 10^{-4}$ | 301s |
| ringnorm | $0.736 \pm 10^{-3}$ | 1228s | $\mathbf{0.741} \pm 6 \cdot 10^{-4}$ | 82s |
| banana | $0.55 \pm 10^{-2}$ | 1229s | $\mathbf{0.559} \pm 10^{-2}$ | 66s |
| splice | $0.847 \pm 2 \cdot 10^{-3}$ | 2290s | $\mathbf{0.85} \pm 2 \cdot 10^{-3}$ | 113s |
| waveform | $\mathbf{0.782} \pm 8 \cdot 10^{-4}$ | 856s | $0.776 \pm 8 \cdot 10^{-4}$ | 120s |
| image | $\mathbf{0.822} \pm 10^{-3}$ | 1947s | $0.821 \pm 3 \cdot 10^{-3}$ | 72s |

also report the training time and see that SWGFs allow to obtain results as good as with JKO-ICNN for most of the datasets but for shorter training times which underlines the better complexity of our scheme.

## 4.2 Convergence to Stationary Distribution for an Aggregation Equation

We also show the possibility to find the stationary solution of different PDEs than Fokker-Planck. For example, using an interaction functional of the form

$$\mathcal{W}(\mu) = \frac{1}{2} \iint W(x - y) \, \mathrm{d}\mu(x)\mathrm{d}\mu(y). \tag{25}$$

We notice here that we do not need to evaluate the density. Therefore, we can apply any neural network. For example, in the following, we will use a simple fully connected neural network (FCNN) and compare the results obtained with JKO-ICNN. We also show the results when learning directly over the particles and when learning weights over a regular grid.

Carrillo et al. (2021) use a repulsive-attractive interaction potential $W(x) = \frac{\|x\|_2^4}{4} - \frac{\|x\|_2^2}{2}$. In this case, they showed empirically that the solution is a Dirac ring with radius 0.5 and centered at the origin when starting from $\mu_0 = \mathcal{N}(0, 0.25^2 I_2)$. With $\tau = 0.05$, we show on Figure 4 that we recover this result with SWGFs for different parametrizations of the probabilities. More precisely, we first use a discretized grid of $50 \times 50$ samples of $[-1, 1]^2$. Then, we show the results when directly learning the particles and when using a FCNN. We also compare with the results obtained with JKO-ICNN. The densities reported for the last three methods are obtained through a kernel density estimator (KDE) with a bandwidth manually chosen since we either do not have access to the density, or we observed for JKO-ICNN that the likelihood exploded (see Appendix D.4). It may be due to the fact that the stationary solution does not admit a density with respect to the Lebesgue measure. For JKO-ICNN, we observe that the ring shape is recovered, but the samples are not evenly distributed on it.

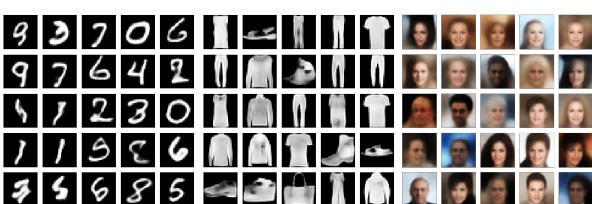

Figure 5: Generated sample obtained through a pre-trained decoder + RealNVP.

Table 2: FID scores on some datasets (lower is better)

| | Methods | MNIST | Fashion | CelebA |
|---|---|---|---|---|
| Ambient Space | SWF (Liutkus et al., 2019) | 225.1 | 207.6 | - |
| | SWGF + RealNVP | 88.1 | **95.5** | - |
| | SWGF + CNN | **69.3** | 102.3 | - |
| Latent Space | AE (golden score) | 15.55 | 31 | 77 |
| | SWGF + AE + RealNVP | **17.8** | **40.6** | 90.9 |
| | SWGF + AE + FCNN | 18.3 | 41.7 | **88** |
| | SWF | 22.5 | 56.4 | 91.2 |

We report the solution at time $t = 10$, and used $\tau = 0.05$ for SW-JKO and $\tau = 0.1$ for JKO-ICNN. As JKO-ICNN requires $O(k^2)$ evaluations of gradients of ICNNs, the training is very long for such a dynamic. Here, the training took around 5 hours on a RTX 2080 TI (for 100 steps), versus 20 minutes for the FCNN and 10 minutes for 1000 particles (for 200 steps).

This underlines again the better training complexity of SW-JKO compared to JKO-ICNN, which is especially appealing when we are only interested in learning the optimal distribution. One such task is generative modeling in which we are interested in learning a target distribution $\nu$ from which we have access through samples.

### 4.3 Application on Real Data

In what follows, we show that the SW-JKO scheme can generate real data, and perform better than the associated particle scheme. To perform generative modeling, we can use different functionals. For example, GANs use the Jensen-Shannon divergence (Goodfellow et al., 2014) and WGANs the Wasserstein-1 distance (Arjovsky et al., 2017). To compare with an associated particle scheme, we focus here on the regularized SW distance as functional, defined as

$$\mathcal{F}(\mu) = \frac{1}{2}SW_2^2(\mu, \nu) + \lambda\mathcal{H}(\mu), \tag{26}$$

where $\nu$ is some target distribution, for which we should have access to samples. The Wasserstein gradient flow of this functional was first introduced and study by Bonnotte (2013) for $\lambda = 0$, and by Liutkus et al. (2019) with the negative entropy term. Liutkus et al. (2019) showcased a particle scheme called SWF (Sliced Wasserstein Flow) to approximate the WGF of equation 26. Applied on images such as MNIST (LeCun & Cortes, 2010), FashionMNIST (Xiao et al., 2017) or CelebA (Liu et al., 2015), SWFs need a very long convergence due to the curse of dimensionality and the trouble approximating SW. Hence, they used instead a pretrained autoencoder (AE) and applied the particle scheme in the latent space. Likewise, we use the AE proposed by Liutkus et al. (2019) with a latent space of dimension $d = 48$, and we perform SW-JKO steps on thoses images. We report on Figure 5 samples obtained with RealNVPs and on Table 2 the Fréchet Inception distance (FID) (Heusel et al., 2017) obtained between $10^4$ samples. We denote "golden score" the FID obtained with the pretrained autoencoder. Hence, we cannot obtain better results than this. We compared the results in the latent and in the ambient space with SWFs and see that we obtain fairly better results using generative models within the SW-JKO scheme, especially in the ambient space, although the results are not really competitive with state-of-the-art methods. This may be due more to the curse of dimensionality in approximating the objective SW than in approximating the regularizer SW.

To sum up, an advantage of the SW-JKO scheme is to be able to use easier, yet powerful enough, architectures to learn the dynamic. This is cheaper in training time and less memory costly. Furthermore, we can tune the architecture with respect to the characteristics of the problem and add inductive biases (*e.g.* using CNN for images) or learn directly over the particles for low dimensional problems.

## 5 Conclusion

In this work, we derive a new class of gradient flows in the space of probability measures endowed with the sliced-Wasserstein metric, and the corresponding algorithms. To the best of our knowledge, and despite its

simplicity, this is the first time that this class of flows is proposed in a machine learning context. We showed that it has several advantages over state-of-the-art approaches such as the recent JKO-ICNN. Aside from being less computationally intensive, it is more versatile *w.r.t.* the different practical solutions for modeling probability distributions, such as normalizing flows, generative models or sets of evolving particles.

Regarding the theoretical aspects, several challenges remain ahead: First, its connections with Wasserstein gradient flows are still unclear. Second, one needs to understand if, regarding the optimization task, convergence speeds or guarantees are changed with this novel formulation, revealing potentially interesting practical properties. Lastly, it is natural to study if popular variants of the sliced-Wasserstein distance such as Max-sliced (Deshpande et al., 2019), Distributional sliced (Nguyen et al., 2021), Subspace robust (Paty & Cuturi, 2019), generalized sliced (Kolouri et al., 2019) or projection Wasserstein distances (Rowland et al., 2019) can also be used in similar gradient flow schemes. The study of higher-order approximation schemes such as BDF2 (Matthes & Plazotta, 2019; Plazotta, 2018) could also be of interest.

### Acknowledgments

This research was funded by project DynaLearn from Labex CominLabs and Région Bretagne ARED DLearnMe, and by the project OTTOPIA ANR-20-CHIA-0030 of the French National Research Agency (ANR).

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

# A    Some Results of (Candau-Tilh, 2020)

In this section, we report some interesting theoretical results of (Candau-Tilh, 2020). More precisely, in Appendix A.1, we report the geodesics in SW space. Then, in Appendix A.2, we report the Euler-Lagrange equation of the Wasserstein gradient flow and the Sliced-Wasserstein gradient flow of the Fokker-Planck functional. Finally, in Appendix A.3, we report some theoretical results that we will use in our proofs.

## A.1    Geodesics in Sliced-Wasserstein Space

We recall here first some notions in metric spaces. Let $(X, d)$ be some metric space. In our case, we will have $X = \mathcal{P}_2(\Omega)$ with $\Omega$ a bounded, open convex set and $d = SW_2$.

We first need to define an absolutely continuous curve.

**Definition 1** (Absolutely continuous curve). *A curve $w : [0,1] \to X$ is said to be absolutely continuous if there exists $g \in L^1([0,1])$ such that*

$$\forall t_0 < t_1, \ d\big(w(t_0), w(t_1)\big) \leq \int_{t_0}^{t_1} g(s)\mathrm{d}s. \tag{27}$$

We denote by $AC(X, d)$ the set of absolutely continuous measures. Moreover, denote $AC_{x,y}(X, d)$ the set of curves in $AC(X, d)$ starting at $x$ and ending at $y$. Then, we can define the length of an absolutely continuous curve $w \in AC(X, d)$ as

$$L_d(w) = \sup\left\{\sum_{k=0}^{n-1} d\big(w(t_k), w(t_{k+1})\big), \ n \geq 1, \ 0 = t_0 < t_1 < \cdots < t_n = 1\right\}. \tag{28}$$

Then, we say that a space $X$ is a geodesic space if for any $x, y \in X$,

$$d(x, y) = \min\left\{L_d(w), \ w \in AC(X, d), w(0) = x, w(1) = y\right\}. \tag{29}$$

Finally, Candau-Tilh (2020) showed in Theorem 2.4 that $(\mathcal{P}_2(\Omega), SW_2)$ is not a geodesic space but rather a pseudo-geodesic space since for $\mu, \nu \in \mathcal{P}_2(\Omega)$,

$$\inf\left\{L_{SW_2}(w), \ w \in AC_{\mu,\nu}(\mathcal{P}_2(\Omega), SW_2)\right\} = c_{d,2} W_2(\mu, \nu). \tag{30}$$

We see that the infimum of the length in the SW space is the Wasserstein distance. Hence, it suggests that the geodesics in SW space are related to the one in Wasserstein space, which are well known since they correspond to the Mc Cann's interpolation (see *e.g.* (Santambrogio, 2015, Theorem .27)).

## A.2    Euler-Lagrange Equations in Wasserstein and Sliced-Wasserstein Spaces

To prove that the Wasserstein gradient flow of a functional satisfies a PDE, one step consists at computing the first variations of the functional in the JKO scheme, *i.e.* of $x \mapsto d^2(x, y)/2\tau + F(x)$. Evaluating the first variation at the minimizer actually gives a corresponding Euler-Lagrange equation (Santambrogio, 2015; Candau-Tilh, 2020).

Hence, Candau-Tilh (2020) also provided a similar form of Euler-Lagrange equations for Wasserstein and Sliced-Wasserstein gradient flows of the Fokker-Planck functional

$$\mathcal{F}(\mu) = \int V\mathrm{d}\mu + \mathcal{H}(\mu). \tag{31}$$

**Proposition 3** (Proposition 3.9 in (Candau-Tilh, 2020)).

- *Let $\mu_{k+1}^\tau$ be the optimal measure in*

$$\underset{\mu \in \mathcal{P}_2(K)}{\arg\min} \ \frac{W_2^2(\mu, \mu_k^\tau)}{2\tau} + \mathcal{F}(\mu). \tag{32}$$

  *Then, it satisfies the following Euler-Lagrange equation:*

$$\log(\rho_{k+1}^\tau) + V + \frac{\psi}{\tau} = \text{constant a.e.,} \tag{33}$$

  *where $\rho_{k+1}^\tau$ is the density of $\mu_{k+1}^\tau$ and $\psi$ is the Kantorovitch potential from $\mu_{k+1}^\tau$ to $\mu_k^\tau$.*

- *Let $\mu_{k+1}^\tau$ be the optimal measure in*

$$\underset{\mu \in \mathcal{P}_2(K)}{\arg\min} \ \frac{SW_2^2(\mu, \mu_k^\tau)}{2\tau} + \mathcal{F}(\mu). \tag{34}$$

  *Then, it satisfies the following Euler-Lagrange equation:*

$$\log(\rho_{k+1}^\tau) + V + \frac{1}{\tau} \int_{S^{d-1}} \psi_\theta \circ P^\theta \ \mathrm{d}\lambda(\theta) = \text{constant a.e.,} \tag{35}$$

  *where for $\theta \in S^{d-1}$, $\psi_\theta$ is the Kantorovitch potential form $P_\#^\theta \mu_{k+1}^\tau$ to $P_\#^\theta \mu_k^\tau$.*

From this proposition, we see that the Euler-Lagrange equations share similar forms, since only the first variation of the distance changes.

Informally, one method to find the corresponding PDE for Wasserstein gradient flows is to show that the derivative of the first variation of Wasserstein is equal (weakly, *i.e.* in the sense of distributions by integrating *w.r.t.* test functions) to $\frac{\partial \rho}{\partial t}$. In this case, we recover well from (32) the PDE. Indeed, $\rho$ satisfies the PDE

$$\frac{\partial \rho}{\partial t} = \text{div}(\rho \nabla V) + \Delta \rho \tag{36}$$

in a weak sense if for all $\xi \in C^\infty(]0, +\infty[\times \mathbb{R}^d)$,

$$\int_0^{+\infty} \int_{\mathbb{R}^d} \left( \frac{\partial \xi}{\partial t}(t,x) + \langle \nabla V(x), \nabla_x \xi(t,x) \rangle - \Delta \xi(t,x) \right) \ \mathrm{d}\rho_t(x)\mathrm{d}t = -\int \xi(0,x) \ \mathrm{d}\rho_0(x). \tag{37}$$

By using the Euler-Lagrange equation (33), informally, we have $\rho \nabla \left( \log(\rho) + V + \frac{\psi}{\tau} \right) = \nabla \rho + \rho \nabla V + \rho \frac{\nabla \psi}{\tau}$. Then, we integrate $\nabla \xi$ *w.r.t.* to this distribution and we use that (by integration by parts)

$$\int \langle \nabla \xi(x), \nabla \rho(x) \rangle \mathrm{d}x = -\int \Delta \xi(x) \ \mathrm{d}\rho(x), \tag{38}$$

and we obtain

$$\int_{\mathbb{R}^d} \left( -\Delta \xi(x) + \langle \nabla \xi(x), \nabla V(x) \rangle + \frac{1}{\tau} \langle \nabla \xi(x), \nabla \psi(x) \rangle \right) \ \mathrm{d}\rho_t(x) = 0. \tag{39}$$

Hence, to obtain the right equation, we require to show that in the limit $\tau \to 0$, the term containing the first variation of Wasserstein is equal to the term representing the derivation in time, *i.e.*

$$\lim_{\tau \to 0} \ \frac{1}{\tau} \iint \langle \nabla \xi(x), \nabla \psi(x) \rangle \mathrm{d}\rho_t(x)\mathrm{d}t = \iint \frac{\partial \xi}{\partial t} \ \mathrm{d}\rho_t(x)\mathrm{d}t. \tag{40}$$

This is done *e.g.* in (Bonnotte, 2013, Theorem 5.3.6) or (Liutkus et al., 2019, Theorem S6). For the Sliced-Wasserstein distance, it is still an open question whether we have the same connection or not.

Note that we recover the divergence operator by using an integration by parts (see *e.g.* (Korba et al., 2021, Appendix A.1))

$$\int \langle \nabla \xi(x), \nabla V(x) \rangle \mathrm{d}\rho(x) = -\int \xi(x) \text{div}\big(\rho(x) \nabla V(x)\big) \ \mathrm{d}x. \tag{41}$$

For a complete derivation of the Wasserstein gradient flows of the Fokker-Planck functional, see *e.g.* (Santambrogio, 2015, Section 4.5).

### A.3 Results on the SW-JKO Scheme

Finally, we report here some results about the continuity and convexity of the Sliced-Wasserstein distance as well as on the existence of the minimizer at each step of the SW-JKO scheme. We will use these results in our proofs in Appendix B.

In the following, we restrain ourselves to measures supported on a compact domain $K$.

**Proposition 4** (Proposition 3.4 in (Candau-Tilh, 2020)). *Let $\nu \in \mathcal{P}_2(K)$. Then, $\mu \mapsto SW_2^2(\mu, \nu)$ is continuous* w.r.t. *the weak convergence.*

**Proposition 5** (Proposition 3.5 in (Candau-Tilh, 2020)). *Let $\nu \in \mathcal{P}_2(K)$, then $\mu \mapsto SW_2^2(\mu, \nu)$ is convex and stricly convex whenever $\nu$ is absolutely continuous* w.r.t. *the Lebesgue measure.*

**Proposition 6** (Proposition 3.7 in (Candau-Tilh, 2020)). *Let $\tau > 0$ and $\mu_k^\tau \in \mathcal{P}_2(K)$. Then, there exists a unique solution $\mu_{k+1}^\tau \in \mathcal{P}_2(K)$ to the minimization problem*

$$\min_{\mu \in \mathcal{P}_2(K)} \frac{SW_2^2(\mu, \mu_k^\tau)}{2\tau} + \int V \, \mathrm{d}\mu + \mathcal{H}(\mu). \tag{42}$$

*The solution is even absolutely continuous.*

## B Proofs

### B.1 Proof of Proposition 1

We refer to the proposition 3.7 in (Candau-Tilh, 2020).

Let $\tau > 0$, $k \in \mathbb{N}$, $\mu_k^\tau \in \mathcal{P}_2(K)$. Let's note $J(\mu) = \frac{SW_2^2(\mu, \mu_k^\tau)}{2\tau} + \mathcal{F}(\mu)$.

According to Proposition 3.4 in (Candau-Tilh, 2020), $\mu \mapsto SW_2^2(\mu, \mu_k^\tau)$ is continuous with respect to the weak convergence. Indeed, let $\mu \in \mathcal{P}_2(K)$ and let $(\mu_n)_n$ converging weakly to $\mu$, *i.e.* $\mu_n \xrightarrow[n \to \infty]{\mathcal{L}} \mu$. Then, by the reverse triangular inequality, we have

$$|SW_2(\mu_n, \mu_k^\tau) - SW_2(\mu, \mu_k^\tau)| \le SW_2(\mu_n, \mu) \le W_2(\mu_n, \mu). \tag{43}$$

Since the Wasserstein distance metrizes the weak convergence (Villani, 2008), we have that $W_2(\mu_n, \mu) \to 0$. And therefore, $\mu \mapsto SW_2(\mu, \mu_k^\tau)$ and $\mu \mapsto SW_2^2(\mu, \mu_k^\tau)$ are continuous *w.r.t.* the weak convergence.

By hypothesis, $\mathcal{F}$ is lower semi continuous, hence $\mu \mapsto J(\mu)$ is lower semi continuous. Moreover, $\mathcal{P}_2(K)$ is compact for the weak convergence, thus we can apply the Weierstrass theorem (Box 1.1 in (Santambrogio, 2015)) and there exists a minimizer $\mu_{k+1}^\tau$ of $J$.

By Proposition 3.5 in (Candau-Tilh, 2020), $\mu \mapsto SW_2^2(\mu, \nu)$ is convex and strictly convex whenever $\nu$ is absolutely continuous *w.r.t.* the Lebesgue measure. Hence, for the uniqueness, if $\mathcal{F}$ is strictly convex then $\mu \mapsto J(\mu)$ is also strictly convex and the minimizer is unique. And if $\rho_k^\tau$ is absolutely continuous, then according to Proposition 3.5 in (Candau-Tilh, 2020), $\mu \mapsto SW_2^2(\mu, \mu_k^\tau)$ is strictly convex, and hence $\mu \mapsto J(\mu)$ is also strictly convex since $\mathcal{F}$ was taken convex by hypothesis.

### B.2 Proof of Proposition 2

Let $k \in \mathbb{N}$, then since $\mu_{k+1}^\tau$ is the minimizer of equation 17,

$$\mathcal{F}(\mu_{k+1}^\tau) + \frac{SW_2^2(\mu_{k+1}^\tau, \mu_k^\tau)}{2\tau} \le \mathcal{F}(\mu_k^\tau) + \frac{SW_2^2(\mu_k^\tau, \mu_k^\tau)}{2\tau} = \mathcal{F}(\mu_k^\tau). \tag{44}$$

Hence, as $SW_2^2(\mu_{k+1}^\tau, \mu_k^\tau) \ge 0$,

$$\mathcal{F}(\mu_{k+1}^\tau) \le \mathcal{F}(\mu_k^\tau). \tag{45}$$

### B.3 Upper Bound on the Errors

Following Hwang et al. (2021), we can also derive an upper bound on the error made at each step.

**Proposition 7.** *Let $k \in \mathbb{N}$, $\mu_0 \in \mathcal{P}_2(\mathbb{R}^d)$, $C$ some constant, and assume that $\mathcal{F}$ admits a lower bound, then*

$$
\begin{aligned}
SW_2\big((g_\theta^{k+1,\tau})_{\#}p_Z, \mu_{k+1}^\tau\big) &\leq SW_2\big((g_\theta^{k,\tau})_{\#}p_Z, \mu_k^\tau\big) + C\tau^{\frac{1}{2}} \\
&\leq (k+1)C\tau^{\frac{1}{2}} + SW_2(\mu_0, \mu_1^\tau).
\end{aligned}
\tag{46}
$$

*Proof.* The bound can be found by applying Theorem 3.1 in (Hwang et al., 2021) with $X = (\mathcal{P}_2(\mathbb{R}^d, SW_2)$, and then by applying a straightforward induction. We report here the proof for the sake of completeness.

Let $k \in \mathbb{N}$, then since $\mu_{k+1}^\tau$ is the minimizer of equation 17,

$$
\mathcal{F}(\mu_{k+1}^\tau) + \frac{SW_2^2(\mu_{k+1}^\tau, \mu_k^\tau)}{2\tau} \leq \mathcal{F}(\mu_k^\tau) + \frac{SW_2^2(\mu_k^\tau, \mu_k^\tau)}{2\tau} = \mathcal{F}(\mu_k^\tau).
\tag{47}
$$

Therefore, using that $\mathcal{F}$ is non increasing along $(\mu_k^\tau)_k$ (Proposition 2), we have $\mathcal{F}(\mu_k^\tau) \leq \mathcal{F}(\mu_0)$. Moreover, using that $\mathcal{F}$ admits an infimum, we find

$$
SW_2^2(\mu_k^\tau, \mu_{k+1}^\tau) \leq 2\tau\big(\mathcal{F}(\mu_k^\tau) - \mathcal{F}(\mu_{k+1}^\tau)\big) \leq 2\tau\big(\mathcal{F}(\mu_0) - \inf_\mu \mathcal{F}(\mu)\big).
\tag{48}
$$

Let $A = \mathcal{F}(\mu_0) - \inf_\mu \mathcal{F}(\mu)$. By the same reasoning, we have that

$$
SW_2^2\big((g_\theta^{k+1,\tau})_{\#}p_Z, (g_\theta^{k,\tau})_{\#}p_Z\big) \leq 2\tau A.
\tag{49}
$$

Now, using the triangular inequality, we have that

$$
\begin{aligned}
SW_2\big((g_\theta^{k+1,\tau})_{\#}p_Z, \mu_{k+1}^\tau\big) &\leq SW_2\big((g_\theta^{k+1,\tau})_{\#}p_Z, (g_\theta^k)_{\#}p_Z\big) + SW_2\big((g_\theta^k)_{\#}p_Z, \mu_k^\tau\big) + SW_2(\mu_k^\tau, \mu_{k+1}^\tau) \\
&= 2\sqrt{2\tau A} + SW_2\big((g_\theta^{k,\tau})_{\#}p_Z, \mu_k^\tau\big).
\end{aligned}
\tag{50}
$$

Let $C = 2\sqrt{2A}$, then by induction we find

$$
SW_2\big((g_\theta^{k+1,\tau})_{\#}p_Z, \mu_{k+1}^\tau\big) \leq (k+1)C\sqrt{\tau} + SW_2(\mu_0, \mu_1^\tau).
\tag{51}
$$

$\square$

### B.4 Sliced-Wasserstein results

**Link for 1D supported measures.** Let $\mu, \nu \in \mathcal{P}(\mathbb{R}^d)$ supported on a line. For simplicity, we suppose that the measures are supported on an axis, *i.e.* $\mu(x) = \mu_1(x_1) \prod_{i=2}^d \delta_0(x_i)$ and $\nu(x) = \nu_1(x_1) \prod_{i=2}^d \delta_0(x_i)$.

In this case, we have that

$$
W_2^2(\mu, \nu) = W_2^2(P_{\#}^{e_1}\mu, P_{\#}^{e_1}\nu) = \int_0^1 |F_{P_{\#}^{e_1}\mu}^{-1}(x) - F_{P_{\#}^{e_1}\nu}^{-1}(x)|^2 \, \mathrm{d}x.
\tag{52}
$$

On the other hand, let $\theta \in S^{d-1}$, then we have

$$
\begin{aligned}
\forall y \in \mathbb{R}, \ F_{P_{\#}^\theta\mu}(y) &= \int_{\mathbb{R}} \mathbb{1}_{]-\infty, y]}(x) \ P_{\#}^\theta\mu(\mathrm{d}x) \\
&= \int_{\mathbb{R}^d} \mathbb{1}_{]-\infty, y]}(\langle \theta, x \rangle) \ \mu(\mathrm{d}x) \\
&= \int_{\mathbb{R}} \mathbb{1}_{]-\infty, y]}(x_1 \theta_1) \ \mu_1(\mathrm{d}x_1) \\
&= \int_{\mathbb{R}} \mathbb{1}_{]-\infty, \frac{y}{\theta_1}]}(x_1) \ \mu_1(\mathrm{d}x_1) \\
&= F_{P_{\#}^{e_1}\mu}\left(\frac{y}{\theta_1}\right).
\end{aligned}
\tag{53}
$$

Therefore, $F_{P_\#^\theta \mu}^{-1}(z) = \theta_1 F_{P_\#^{e_1}\mu}^{-1}(z)$ and

$$
\begin{aligned}
W_2^2(P_\#^\theta \mu, P_\#^\theta \nu) &= \int_0^1 |\theta_1 F_{P_\#^{e_1}\mu}^{-1}(z) - \theta_1 F_{P_\#^{e_1}\nu}^{-1}(z)|^2 \ \mathrm{d}z \\
&= \theta_1^2 \int_0^1 |F_{P_\#^{e_1}\mu}^{-1}(z) - F_{P_\#^{e_1}\nu}^{-1}(z)|^2 \ \mathrm{d}z \\
&= \theta_1^2 W_2^2(\mu, \nu).
\end{aligned} \tag{54}
$$

Finally, using that $\int_{S^{d-1}} \theta\theta^T \mathrm{d}\lambda(\theta) = \frac{1}{d}I_d$, we can conclude that

$$
SW_2^2(\mu, \nu) = \int_{S^{d-1}} \theta_1^2 W_2^2(\mu, \nu)\mathrm{d}\theta = \frac{W_2^2(\mu, \nu)}{d}.
$$

**Closed-form between Gaussians.** It is well known that there is a closed-form for the Wasserstein distance between Gaussians (Givens & Shortt, 1984). If we take $\alpha = \mathcal{N}(\mu, \Sigma)$ and $\beta = \mathcal{N}(m, \Lambda)$ with $m, \mu \in \mathbb{R}^d$ and $\Sigma, \Lambda \in \mathbb{R}^{d \times d}$ two symmetric positive definite matrices, then

$$
W_2^2(\alpha, \beta) = \|m - \mu\|_2^2 + \mathrm{Tr}\big(\Sigma + \Lambda - 2(\Sigma^{\frac{1}{2}} \Lambda \Sigma^{\frac{1}{2}})^{\frac{1}{2}}\big). \tag{55}
$$

Let $\alpha = \mathcal{N}(\mu, \sigma^2 I_d)$ and $\beta = \mathcal{N}(m, s^2 I_d)$ two isotropic Gaussians. Here, we have

$$
\begin{aligned}
W_2^2(\alpha, \beta) &= \|\mu - m\|_2^2 + \mathrm{Tr}(\sigma^2 I_d + s^2 I_d - 2(\sigma s^2 \sigma I_d)^{\frac{1}{2}}) \\
&= \|\mu - m\|_2^2 + (\sigma - s)^2 \ \mathrm{Tr}(I_d) \\
&= \|\mu - m\|_2^2 + d(\sigma - s)^2.
\end{aligned} \tag{56}
$$

On the other hand, Nadjahi et al. (2021) showed (Equation 73) that

$$
SW_2^2(\alpha, \beta) = \frac{\|\mu - m\|_2^2}{d} + (\sigma - s)^2 = \frac{W_2^2(\alpha, \beta)}{d}. \tag{57}
$$

In that case, the dilation of factor $d$ between WGF and SWGF clearly appears.

For more complicated gaussians, we may not have this equality. For example, let $\alpha = \mathcal{N}(\mu, D)$, $\beta = \mathcal{N}(m, \Delta)$ with $D$ and $\Delta$ diagonal. Then, $P_\#^\theta \alpha = \mathcal{N}(\langle \mu, \theta \rangle, \sum_{i=1}^N \theta_i^2 D_i)$, $P_\#^\theta \beta = \mathcal{N}(\langle m, \theta \rangle, \sum_{i=1}^N \theta_i^2 \Delta_i)$ and

$$
W_2^2(P_\#^\theta \bar\alpha, P_\#^\theta \bar\beta) = \left(\sqrt{\sum_i \theta_i^2 D_i} - \sqrt{\sum_i \theta_i^2 \Delta_i}\right)^2 \tag{58}
$$

with $\bar\alpha, \bar\beta$ the centered measures (noting $T^\alpha : x \mapsto x - \mu$, then $\bar\alpha = T_\#^\alpha \alpha$). Hence, we have

$$
\begin{aligned}
SW_2^2(\alpha, \beta) &= \frac{\|\mu - m\|_2^2}{d} + SW_2^2(\bar\alpha, \bar\beta) \\
&= \frac{\|\mu - m\|_2^2}{d} + \int_{S^{d-1}} \left(\sqrt{\sum_i \theta_i^2 D_i} - \sqrt{\sum_i \theta_i^2 \Delta_i}\right)^2 \ \mathrm{d}\lambda(\theta) \\
&= \frac{\|\mu - m\|_2^2}{d} + \int_{S^{d-1}} \left(\sum_{i=1}^d \theta_i^2 D_i + \sum_{i=1}^d \theta_i^2 \Delta_i - 2\sqrt{\sum_{i,j} \theta_i^2 \theta_j^2 D_i \Delta_j}\right)^2 \ \mathrm{d}\lambda(\theta) \\
&= \frac{\|\mu - m\|_2^2}{d} + \sum_{i=1}^d D_i \int_{S^{d-1}} \theta_i^2 \mathrm{d}\lambda(\theta) + \sum_{i=1}^D \Delta_i \int_{S^{d-1}} \theta_i^2 \mathrm{d}\lambda(\theta) - 2 \int_{S^{d-1}} \sqrt{\sum_{i,j} \theta_i^2 \theta_j^2 D_i \Delta_j} \mathrm{d}\lambda(\theta) \\
&= \frac{\|\mu - m\|_2^2}{d} + \frac{1}{d} \sum_{i=1}^d (D_i + \Delta_i) - 2 \int_{S^{d-1}} \sqrt{\sum_{i,j} \theta_i^2 \theta_j^2 D_i \Delta_j} \mathrm{d}\lambda(\theta),
\end{aligned} \tag{59}
$$

using that $\int_{S^{d-1}} \theta\theta^T \mathrm{d}\lambda(\theta) = \frac{1}{d}I_d$ and by applying Proposition 2 in (Nadjahi et al., 2021) to decompose $SW_2^2(\alpha,\beta) = \frac{\|\mu-m\|_2^2}{d} + SW_2^2(\bar{\alpha},\bar{\beta})$.

On the other hand, we have

$$
\begin{aligned}
W_2^2(\alpha,\beta) &= \|\mu - m\|_2^2 + \mathrm{Tr}\big(D + \Delta - 2(D^{\frac{1}{2}}\Delta D^{\frac{1}{2}})^{\frac{1}{2}}\big) \\
&= \|\mu - m\|_2^2 + \mathrm{Tr}(D + \Delta - 2(D\Delta)^{\frac{1}{2}}) \\
&= \|\mu - m\|_2^2 + \sum_{i=1}^d (D_i + \Delta_i - 2D_i^{\frac{1}{2}}\Delta_i^{\frac{1}{2}}) \\
&= \|\mu - m\|_2^2 + \sum_{i=1}^d (D_i^{\frac{1}{2}} - \Delta_i^{\frac{1}{2}})^2.
\end{aligned}
\tag{60}
$$

Since $SW_2^2(\alpha,\beta) \le \frac{1}{d}W_2^2(\alpha,\beta)$, we have $\sum_{i=1}^d \sqrt{D_i\Delta_i} \le d\int_{S^{d-1}} \sqrt{\sum_{i,j}\theta_i^2\theta_j^2 D_i\Delta_j}\mathrm{d}\lambda(\theta)$.

Let $d=2$, $\sigma,s > 0$ and $D = \mathrm{diag}(\sigma^2, \frac{\sigma^2}{2})$, $\Delta = \mathrm{diag}(\frac{s^2}{2}, s^2)$. In this case, on the one hand, we have

$$
\sum_{i=1}^2 \sqrt{D_i\Delta_i} = \sqrt{2}\sigma s.
\tag{61}
$$

On the other hand,

$$
\begin{aligned}
2\int_{S^1} \sqrt{\sum_{i,j}\theta_i^2\theta_j^2 D_i\Delta_j}\mathrm{d}\lambda(\theta) &= \sqrt{2}\sigma s \int_{S^1} \sqrt{(\theta_1^2+\theta_2^2)^2 + \frac{1}{2}\theta_1^2\theta_2^2}\,\mathrm{d}\lambda(\theta) \\
&= \sqrt{2}\sigma s \int_{S^1\cap\{\theta_1\neq 0,\theta_2\neq 0\}} \sqrt{(\theta_1^2+\theta_2^2)^2 + \frac{1}{2}\theta_1^2\theta_2^2}\,\mathrm{d}\lambda(\theta) \\
&> \sqrt{2}\sigma s \int_{S^1} (\theta_1^2+\theta_2^2)\,\mathrm{d}\lambda(\theta) \\
&= \sqrt{2}\sigma s,
\end{aligned}
\tag{62}
$$

using that $\lambda$ is absolutely continuous with respect to the Lebesgue measure and hence $\lambda(\{\theta_1 = 0\} \cup \{\theta_2 = 0\}) = 0$ and the fact that for every $\theta \in S^1\cap\{\theta_1\neq 0,\theta_2\neq 0\}$, $\sqrt{(\theta_1^2+\theta_2^2)^2 + \frac{1}{2}\theta_1^2\theta_2^2} > \sqrt{(\theta_1^2+\theta_2^2)^2} = \theta_1^2 + \theta_2^2$. From this strict inequality, we deduce that $W_2$ and $d\cdot SW_2$ are not always equal, even in this restricted case. Hence, even in this simple case, we cannot directly conclude that we have a dilation term of $d$ between Wasserstein and Sliced-Wasserstein gradient flows.

## C  Computation of the SW-JKO scheme in practice

### C.1  Approximation of SW

For each inner optimization problem

$$
\mu_{k+1}^\tau \in \arg\min_{\mu\in\mathcal{P}_2(\mathbb{R}^d)} \frac{SW_2^2(\mu,\mu_k^\tau)}{2\tau} + \mathcal{F}(\mu),
\tag{63}
$$

we need to approximate the sliced-Wasserstein distance. To do that, we used Monte-Carlo approximate by sampling $n_\theta$ directions $(\theta_i)_{i=1}^{n_\theta}$ following the uniform distribution on the hypersphere $S^{d-1}$ (which can be done by using the stochastic representation, *i.e.* let $Z \sim \mathcal{N}(0,I_d)$, then $\theta = \frac{Z}{\|Z\|_2} \sim \mathrm{Unif}(S^{d-1})$ (Fang et al., 1992)). Let $\mu,\nu \in \mathcal{P}(\mathbb{R}^d)$, we approximate the sliced-Wasserstein distance as

$$
\widehat{SW}_2^2(\mu,\nu) = \frac{1}{n_\theta}\sum_{i=1}^{n_\theta} W_2^2(P_\#^{\theta_i}\mu, P_\#^{\theta_i}\nu).
\tag{64}
$$

---

**Algorithm 2** SW-JKO with Discrete Grid

---

**Input:** $\mu_0$ the initial distribution with density $\rho_0$, $K$ the number of SW-JKO steps, $\tau$ the step size, $\mathcal{F}$ the functional, $N_e$ the number of epochs to solve each SW-JKO step, $(x_j)_{j=1}^N$ the grid

Let $\rho^{(0)} = \left( \frac{\rho_0(x_j)}{\sum_{\ell=1}^N \rho_0(x_\ell)} \right)_{j=1}^N$

**for** $k = 1$ **to** $K$ **do**

    Initialize the weights $\rho^{(k+1)}$ (with for example a copy of $\rho^{(k)}$)

    // Denote $\mu_{k+1}^\tau = \sum_{j=1}^N \rho_j^{(k+1)} \delta_{x_j}$ and $\mu_k^\tau = \sum_{j=1}^N \rho_j^{(k)} \delta_{x_j}$

    **for** $i = 1$ **to** $N_e$ **do**

        Compute $J(\mu_{k+1}^\tau) = \frac{1}{2\tau} SW_2^2(\mu_k^\tau, \mu_{k+1}^\tau) + \mathcal{F}(\mu_{k+1}^\tau)$

        Backpropagate through $J$ with respect to $\rho^{(k+1)}$

        Perform a gradient step

        Project on the simplex $\rho^{(k+1)}$ using the algorithm of Condat (2016)

    **end for**

**end for**

---

In practice, we compute it for empirical distributions $\hat{\mu}_n$ and $\hat{\nu}_m$, and we approximate the one dimensional Wasserstein distance

$$W_2^2(P_\#^\theta \hat{\mu}_n, P_\#^\theta \hat{\nu}_m) = \int_0^1 |F_{P_\#^\theta \hat{\mu}_n}^{-1}(u) - F_{P_\#^\theta \hat{\nu}_m}^{-1}(u)|^2 \, \mathrm{d}u \tag{65}$$

by the rectangle method.

Overall, the complexity is in $O(n_\theta(n \log n + m \log m))$ where $n$ (resp. $m$) denotes the number of particles of $\hat{\mu}_n$ (resp. $\hat{\nu}_m$).

### C.2 Algorithms to solve the SW-JKO scheme

We provide here the algorithms used to solve the SW-JKO scheme (17) for the discrete grid (Section 3.3) and for the particles (Section 3.3).

**Discrete grid.** We recall that in that case, we model the distributions as $\mu_k^\tau = \sum_{i=1}^N \rho_i^{(k)} \delta_{x_i}$ where we use $N$ samples located at $(x_i)_{i=1}^N$ and $(\rho_i^{(k)})_{i=1}^N$ belongs to the simplex $\Sigma_n$. Hence, the SW-JKO scheme at step $k+1$ rewrites

$$\min_{(\rho_i)_i \in \Sigma_N} \frac{SW_2^2(\sum_{i=1}^N \rho_i \delta_{x_i}, \mu_k^\tau)}{2\tau} + \mathcal{F}(\sum_{i=1}^N \rho_i \delta_{x_i}). \tag{66}$$

We report in Algorithm 2 the whole procedure.

**Particle scheme.** In this case, we model the distributions as empirical distributions and we try to optimize the positions of the particles. Hence, we have $\mu_k^\tau = \frac{1}{N} \sum_{i=1}^N \delta_{x_i^{(k)}}$ and the problem (17) becomes

$$\min_{(x_i)_i} \frac{SW_2^2(\frac{1}{N} \sum_{i=1}^N \delta_{x_i}, \mu_k^\tau)}{2\tau} + \mathcal{F}(\frac{1}{N} \sum_{i=1}^N \delta_{x_i}). \tag{67}$$

In this case, we provide the procedure in Algorithm 3.

## D Additional experiments

### D.1 Dynamic of Sliced-Wasserstein gradient flows

The Fokker-Planck equation (8) is the Wasserstein gradient flow of the functional (6). Moreover, it is well-known to have a counterpart stochastic differential equation (SDE) (see *e.g.* Mackey (1992, Chapter 11)) of

---

**Algorithm 3** SW-JKO with Particles

---

**Input:** $\mu_0$ the initial distribution, $K$ the number of SW-JKO steps, $\tau$ the step size, $\mathcal{F}$ the functional, $N_e$ the number of epochs to solve each SW-JKO step, $N$ the batch size

Sample $(x_j^{(0)})_{j=1}^N \sim \mu_0$ i.i.d

**for** $k = 1$ **to** $K$ **do**

    Initialize $N$ particles $(x_j^{(k+1)})_{j=1}^N$ (with for example a copy of $(x_j^{(k)})_{j=1}^N$)

    // Denote $\mu_{k+1}^\tau = \frac{1}{N}\sum_{j=1}^N \delta_{x_j^{(k+1)}}$ and $\mu_k^\tau = \frac{1}{N}\sum_{j=1}^N \delta_{x_j^{(k)}}$

    **for** $i = 1$ **to** $N_e$ **do**

        Compute $J(\mu_{k+1}^\tau) = \frac{1}{2\tau}SW_2^2(\mu_k^\tau, \mu_{k+1}^\tau) + \mathcal{F}(\mu_{k+1}^\tau)$

        Backpropagate through $J$ with respect to $(x_j^{(k+1)})_{j=1}^N$

        Perform a gradient step

    **end for**

**end for**

---

the form

$$\mathrm{d}X_t = -\nabla V(X_t)\mathrm{d}t + \sqrt{2\beta}\,\mathrm{d}W_t \tag{68}$$

with $(W_t)_t$ a Wiener process. This SDE is actually the well-known Langevin equation. Hence, by approximating it using the Euler-Maruyama scheme, we recover the Unadjusted Langevin Algorithm (ULA) (Roberts & Tweedie, 1996; Wibisono, 2018).

For

$$V(x) = \frac{1}{2}(x-m)^T A(x-m), \tag{69}$$

with $A$ symmetric and definite positive, we obtain an Ornstein-Uhlenbeck process (Le Gall, 2016, Chapter 8). If we choose $\mu_0$ as a Gaussian $\mathcal{N}(m_0, \Sigma_0)$, then we know the Wasserstein gradient flow $\mu_t$ in closed form (Wibisono, 2018; Vatiwutipong & Phewchean, 2019), for all $t > 0$, $\mu_t = \mathcal{N}(m_t, \Sigma_t)$ with

$$\begin{cases} m_t = m + e^{-tA}(m_0 - m) \\ \Sigma_t = e^{-tA}\Sigma_0(e^{-tA})^T + A^{-\frac{1}{2}}(I - e^{-2tA})(A^{-\frac{1}{2}})^T. \end{cases} \tag{70}$$

**Comparison of the evolution of the diffusion between SWGFs and WGFs.** For this experiment, we model the density using RealNVPs (Dinh et al., 2016). More precisely, we use RealNVPs with 5 affine coupling layers, using FCNN for the scaling and shifting networks with 100 hidden units and 5 layers. In both experiments, we always start the scheme with $\mu_0 = \mathcal{N}(0, I)$ and take $n_\theta = 1000$ projections to approximate the sliced-Wasserstein distance. We randomly generate a target Gaussian (using "make_spd_matrix" from scikit-learn (Pedregosa et al., 2011) to generate a random covariance with 42 as seed).

We look at the evolution of the distributions learned between $t = 0$ and $t = 4$ with a time step of $\tau = 0.1$. We compare it with the true Wasserstein gradient flow. On Figure 6a, we observe that they do not seem to match. However, they do converge to the same stationary value. On Figure 6b, we plot the functional along the true WGF dilated of a factor $d = 2$. We see here that the two curves are matching and we observed the same behaviour in higher dimension. Even though we cannot conclude on the PDE followed by SWGFs, this reinforces the conjecture that the SWGF obtained with a step size of $\frac{\tau}{d}$ (*i.e.* using the scheme (18)) is very close to the WGF obtained with a step size of $\tau$. We also report here the evolution of the mean (Fig. 7) and of the variance (Fig. 8). For the mean, it follows as expected the same diffusion. For the variance, it is less clear but it is hard to conclude since there are potentially optimization errors.

**Comparison between JKO-ICNN and SW-JKO.** Following the experiment conducted by Mokrov et al. (2021) in section 4.2, we plot in Figure 9 the symmetric Kullback-Leibler (SymKL) divergence over dimensions between approximated distributions and the true WGF at times $t = 0.5$ and $t = 0.9$. We take the mean over 15 random gaussians (generated using the scikit-learn function (Pedregosa et al., 2011) "make_spd_matrix" for the covariance matrices, and generating the means with a standard normal distribution) for dimensions $d \in \{2, \ldots, 12\}$.

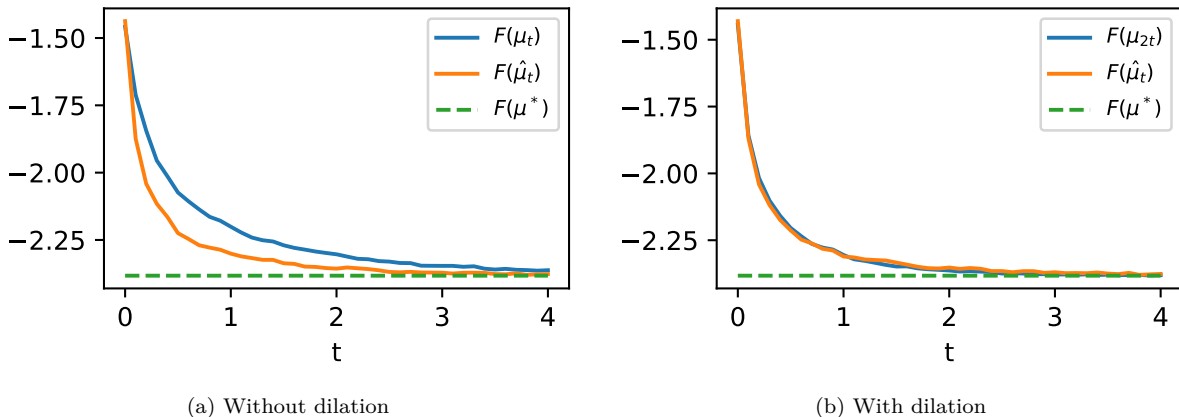

(a) Without dilation
(b) With dilation

Figure 6: Evolution of the functional equation 6 along the WGF $\mu_t$ and the learned SWGF $\hat{\mu}_t$. We observe a dilation of parameter 2 between the WGF and the SWGF.

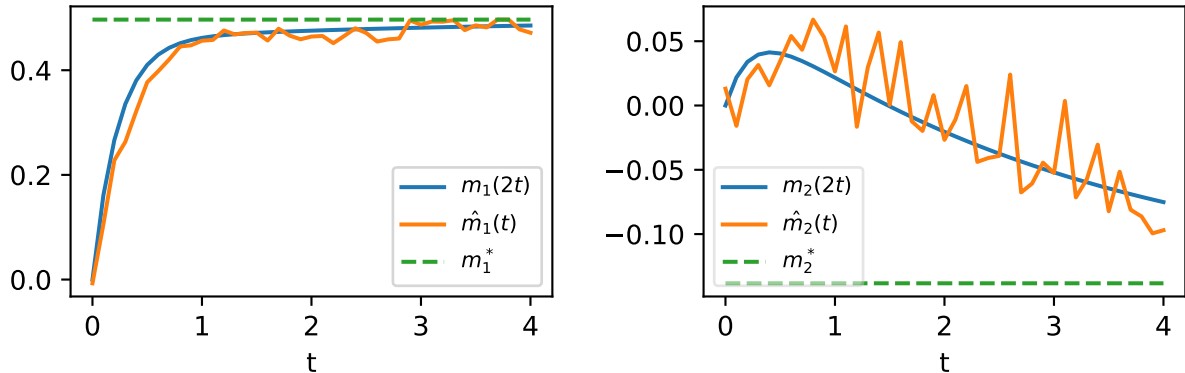

Figure 7: Evolution of the mean taking into account the dilation parameter. $\mu$ denotes the true mean of WGF, $\hat{\mu}$ the mean obtained through SW-JKO (17) with $\tau = 0.05$ and $\mu_*$ the mean of the stationary measure. We observe that the mean of approximated measure obtained through SW-JKO seems to follow the one of the WGF.

For each target Gaussian, we run the SW-JKO dilated scheme (18) with $\tau = 0.05$ for a RealNVP normalizing flow. We compare it with JKO-ICNN with also $\tau = 0.05$ and with Euler-Maruyama with $10^3$, $10^4$ and $5 \cdot 10^4$ particles and a step size of $10^{-3}$. For JKO-ICNN, we use, as Mokrov et al. (2021), DenseICNN with convex quadratic layers introduced in (Korotin et al., 2019) and available at https://github.com/iamalexkorotin/Wasserstein2Barycenters. For the JKO-ICNN scheme, we use our own implementation.

We compute the symmetric Kullback-Leibler divergence between the ground truth of WGF $\mu^*$ and the distribution $\hat{\mu}$ approximated by the different schemes at times $t = 0.5$ and $t = 0.9$. The symmetric Kullback-Leibler divergence is obtained as

$$\mathrm{SymKL}(\mu^*, \hat{\mu}) = \mathrm{KL}(\mu^* || \hat{\mu}) + \mathrm{KL}(\hat{\mu} || \mu^*). \tag{71}$$

To approximate it, we generate $10^4$ samples of each distribution and evaluate the density at those samples.

If we note $g_\theta$ a normalizing flows, $p_Z$ the distribution in the latent space and $\rho = (g_\theta)_\# p_Z$, then we can evaluate the log density of $\rho$ by using the change of variable formula. Let $x = g_\theta(z)$, then

$$\log(\rho(x)) = \log(p_Z(z)) - \log|\det J_{g_\theta}(z)|. \tag{72}$$

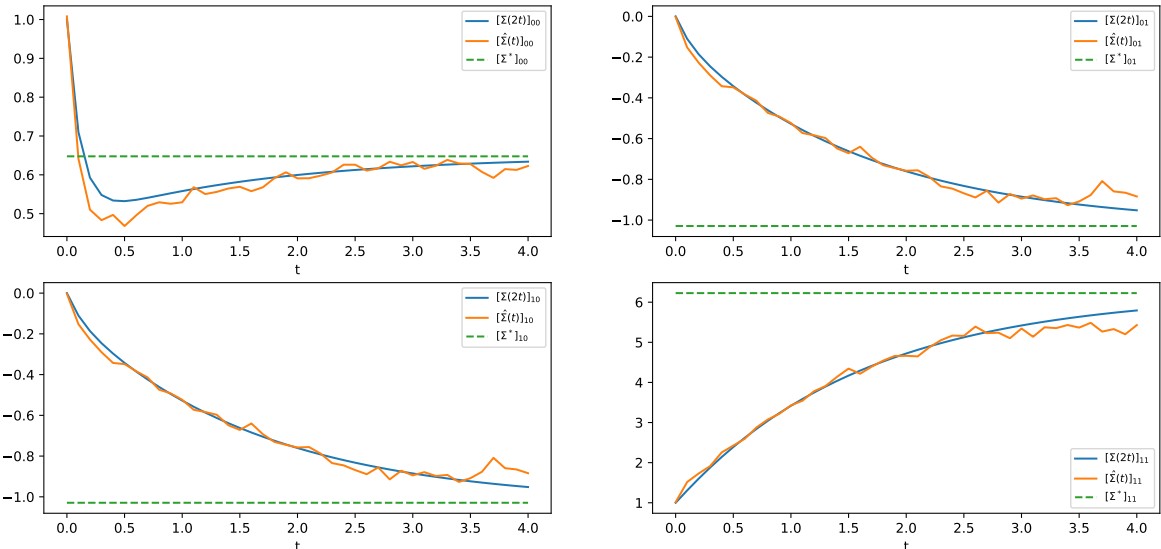

Figure 8: Evolution of the components of the covariance matrix taking into account the dilation parameter. $\Sigma$ denotes the true covariance matrix of WGF, $\hat{\Sigma}$ the covariance matrix obtained through SW-JKO (17) with $\tau = 0.05$ and $\Sigma^*$ the covariance matrix of the stationary distribution. We observe some difference between WGF and SWGF.

We choose RealNVPs (Dinh et al., 2016) for the simplicity of the transformations and the fact that we can compute efficiently the determinant of the Jacobian (since we have a closed-form). A RealNVP flow is a composition of transformations $T$ of the form

$$\forall z \in \mathbb{R}^d, \ x = T(z) = \left(z^1, \exp(s(z^1)) \odot z^2 + t(z^1)\right) \tag{73}$$

where we write $z = (z^1, z^2)$ and with $s$ and $t$ some neural networks. To modify all the components, we use also swap transformations (i.e. $(z^1, z^2) \mapsto (z^2, z^1)$). This transformation is invertible with $\log \det J_T(z) = \sum_i s(z_i^1)$.

For JKO-ICNN, we choose strictly convex ICNNs, and can hence invert them as well as compute the density. In this case, we do not have access to a closed-form for the Jacobian. Therefore, we used backpropagation to compute it. As this experiment is in low dimension, the computational cost is not too heavy. However, there exist stochastic methods to approximate it in greater dimension. We refer to (Huang et al., 2020) and (Alvarez-Melis et al., 2021) for more explanations.

We approximate the functional by using Monte-Carlo approximation as in Section 3.3.

For Euler-Maruyama, as in (Mokrov et al., 2021), we use kernel density estimation in order to approximate the density. We use the scipy implementation (Virtanen et al., 2020) "gaussian_kde" with the Scott's rule to choose the bandwidth.

Finally, we report on the Figure 9 the mean of the log of the symmetric Kullback-Leibler divergence over 15 Gaussians in each dimension and the 95% confidence interval.

For the training of the neural networks, we use an Adam optimizer (Kingma & Ba, 2014) with a learning rate of $10^{-4}$ for RealNVP (except for the 1st iteration where we take a learning rate of $5 \cdot 10^{-3}$) and of $5 \cdot 10^{-3}$ for JKO-ICNN. At each inner optimization step, we start from a deep copy of the last neural network, and optimize RealNVP for 200 epochs and ICNNs for 500 epochs, with a batch size of 1024.

We see on Figure 9 that the results are better than the particle schemes obtained with Euler-Maruyama (EM) with a step size of $10^{-3}$ and with either $10^3$, $10^4$ or $5 \cdot 10^4$ particles in dimension higher than 2. However, JKO-ICNN obtained better results.

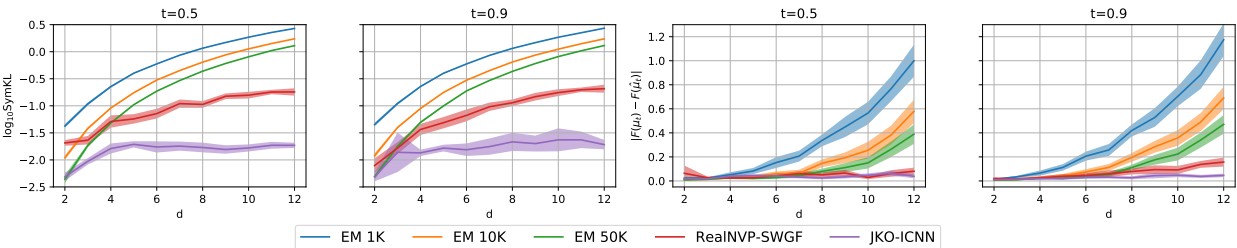

Figure 9: On the left: SymKL divergence at time 0.5 and 0.9 between the groundtruth of the Fokker-Planck equation at time $t$ and the solution of the SW Gradient Flow at time $t$. On the right: Absolute error between the functionals evaluated for WGF and SWGF at time $t$.

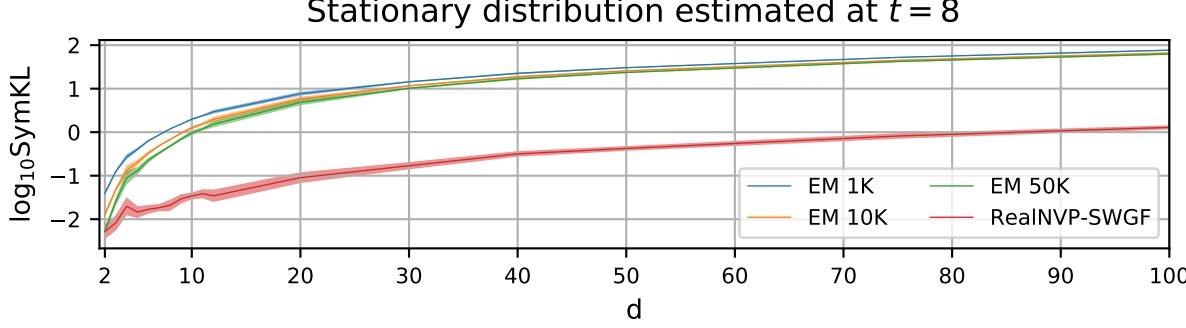

Figure 10: Symmetric KL divergence between the learned distribution at time $t = 8$ and the true stationary solution on Gaussians

### D.2 Convergence to stationary distribution

Here, we want to demonstrate that, through the SW-JKO scheme, we are able to find good minima of functionals using simple generative models.

**Gaussian.** For this experiment, we place ourselves in the same setting of Section 4.1. We start from $\mu_0 = \mathcal{N}(0, I)$ and use a step size of $\tau = 0.1$ for 80 iterations in order to match the stationary distribution. In this case, the functional is

$$\mathcal{F}(\mu) = \int V(x)\mathrm{d}\mu(x) + \mathcal{H}(\mu) \tag{74}$$

with $V(x) = -\frac{1}{2}(x-b)^T A(x-b)$, and the stationary distribution is $\rho^*(x) \propto e^{-V(x)}$, hence $\rho^* = \mathcal{N}(b, A^{-1})$.

We generate 15 Gaussians for $d$ between 2 and 12, and $d \in \{20, 30, 40, 50, 75, 100\}$. Due to the length of the diffusion, and to numerical instabilities, we do not report results obtained with JKO-ICNN. In Figure 2, we showed the results in low dimension (for $d \in \{2, \ldots, 12\}$) and the unstability of JKO-ICNN. We report on Figure 10 the SymKL also in higher dimension.

We use 200 epochs of each inner optimization and an Adam optimizer with a learning rate of $5 \cdot 10^{-3}$ for the first iteration and $10^{-3}$ for the rest. We also use a batch size of 1000 sample.

**Bayesian logistic regression.** For the Bayesian logistic regression, we have access to covariates $s_1, \ldots, s_n \in \mathbb{R}^d$ with their associated labels $y_1, \ldots, y_n \in \{-1, 1\}$. Following (Liu & Wang, 2016; Mokrov et al., 2021), we put as prior on the regression weights $w$, $p_0(w|\alpha) = \mathcal{N}(w; 0, \frac{1}{\alpha})$ with $p_0(\alpha) = \Gamma(\alpha; 1, 0.01)$.

Table 3: Number of features, of samples and batch size of each dataset.

|  | covtype | german | diabetis | twonorm | ringnorm | banana | splice | waveform | image |
|---|---|---|---|---|---|---|---|---|---|
| features | 54 | 20 | 8 | 20 | 20 | 2 | 60 | 21 | 18 |
| samples | 581012 | 1000 | 768 | 7400 | 7400 | 5300 | 2991 | 5000 | 2086 |
| batch size | 512 | 800 | 614 | 1024 | 1024 | 1024 | 512 | 512 | 1024 |

Table 4: Hyperparameters for SWGFs with RealNVPs. nl: number of coupling layers in RealNVP, nh: number of hidden units of conditioner neural networks, lr: learning rate using Adam, JKO steps: number of SW-JKO steps, Iters by step: number of epochs for each SW-JKO step, $\tau$: the time step, batch size: number of samples taken to approximate the functional.

|  | covtype | german | diabetis | twonorm | ringnorm | banana | splice | waveform | image |
|---|---|---|---|---|---|---|---|---|---|
| nl | 2 | 2 | 2 | 2 | 2 | 2 | 5 | 5 | 2 |
| nh | 512 | 512 | 512 | 512 | 512 | 512 | 128 | 128 | 512 |
| lr | $2e^{-5}$ | $1e^{-4}$ | $5e^{-4}$ | $1e^{-4}$ | $5e^{-5}$ | $1e^{-4}$ | $5e^{-4}$ | $1e^{-4}$ | $5e^{-5}$ |
| JKO steps | 5 | 5 | 10 | 20 | 5 | 5 | 5 | 5 | 5 |
| Iters by step | 1000 | 500 | 500 | 500 | 1000 | 500 | 500 | 500 | 500 |
| $\tau$ | 0.1 | $10^{-6}$ | $5 \cdot 10^{-6}$ | $10^{-8}$ | $10^{-6}$ | 0.1 | $10^{-6}$ | $10^{-8}$ | 0.1 |
| batch size | 1024 | 1024 | 1024 | 1024 | 1024 | 1024 | 1024 | 512 | 1024 |

Therefore, we aim at learning the posterior $p(w, \alpha|y)$:

$$p(w, \alpha|y) \propto p(y|w, \alpha)p_0(w|\alpha)p_0(\alpha) = p_0(\alpha)p_0(w|\alpha) \prod_{i=1}^{n} p(y_i|w, \alpha)$$

where $p(y_i|w, \alpha) = \sigma(w^T s_i)^{\frac{1+y_i}{2}}(1 - \sigma(w^T s_i))^{\frac{1-y}{2}}$ with $\sigma$ the sigmoid. To evaluate $\mathcal{V}(\mu) = \int V(x) \, d\mu(x)$, we resample data uniformly.

In our context, let $V(x) = -\log\left(p_0(\alpha)p_0(w|\alpha)p(y|w, \alpha)\right)$, then using $\mathcal{F}(\mu) = \int V d\mu + \mathcal{H}(\mu)$ as functional, we know that the limit of the stationary solution of Fokker-Planck is proportional to $e^{-V} = p(w, \alpha|y)$.

Following Mokrov et al. (2021); Liu & Wang (2016), we use the 8 datasets of Mika et al. (1999) and the covertype dataset (https://www.csie.ntu.edu.tw/~cjlin/libsvmtools/datasets/binary.html).

We report in Table 3 the characteristics of the different datasets. The datasets are loaded using the code of Mokrov et al. (2021) (https://github.com/PetrMokrov/Large-Scale-Wasserstein-Gradient-Flows). We split the dataset between train set and test set with a 4:1 ratio.

We report in Table 4 the hyperparameters used for the results reported in Table 1. We also tuned the time step $\tau$ since for too big $\tau$, we observed bad results, which the SW-JKO scheme should be a good approximation of the SWGF only for small enough $\tau$.

Moreover, we reported in Table 1 the mean over 5 training. For the results obtained with JKO-ICNN, we used the same hyperparameters as Mokrov et al. (2021).

### D.3 Influence of the number of projections

It is well known that the approximation of Sliced-Wasserstein is subject to the curse of dimensionaly through the Monte-Carlo approximation (Nadjahi et al., 2020). We provide here some experiment to quantify this influence. However, first note that the goal is not to minimize the Sliced-Wasserstein distance, but rather the functional, SW playing mostly a regularizer role. Experiments on the influence of the number of experiments to approximate the SW have already been conducted (see *e.g.* Figure 2 in (Nadjahi et al., 2020) or Figure 1 in (Deshpande et al., 2019)).

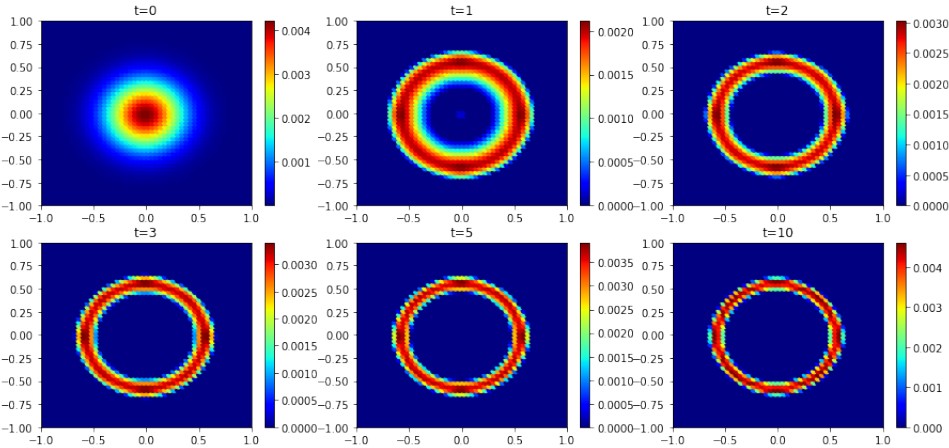

Figure 11: Density over time of the solution of the aggregation equation learned over the discre grid.

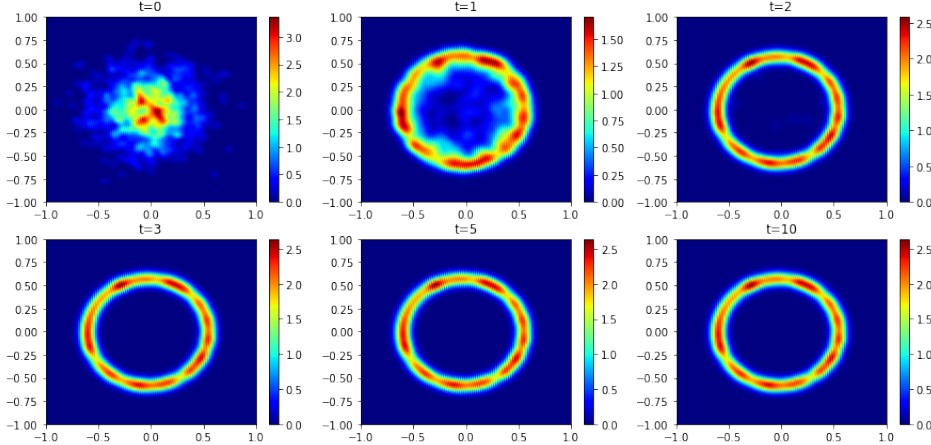

Figure 12: Density over time of the solution of the aggregation equation by learning particles.

Here, we take the same setting of Section 4.1, *i.e.* we generate 15 random Gaussians, and then vary the number of projections and report the Symmetric Kullback-Leibler divergence on Figure 3. We observe that the results seem to improve with the number of projections until it reach a certain plateau. The plateau seems to be attained for a bigger number of dimension in high dimension.

### D.4 Aggregation equations

Here, we use as functional

$$\mathcal{W}(\mu) = \iint W(x - y)\mathrm{d}\mu(x)\mathrm{d}\mu(y). \tag{75}$$

Carrillo et al. (2021) use a repulsive-attractive interaction potential, for $a > b \geq 0$,

$$W(x) = \frac{\|x\|^a}{a} - \frac{\|x\|^b}{b} \tag{76}$$

using the convention $\frac{\|x\|^0}{0} = \ln(\|x\|)$. For some values of $a$ and $b$, there is existence of stable equilibrium state (Balagué et al., 2013).

**Dirac Ring.** First, for the Dirac ring example, we take $a = 4$ and $b = 2$. Then, $W$ is a repulsive-attractive interaction potential (repulsive in the short range, and attractive in the long range because $b < a$ (Balagué

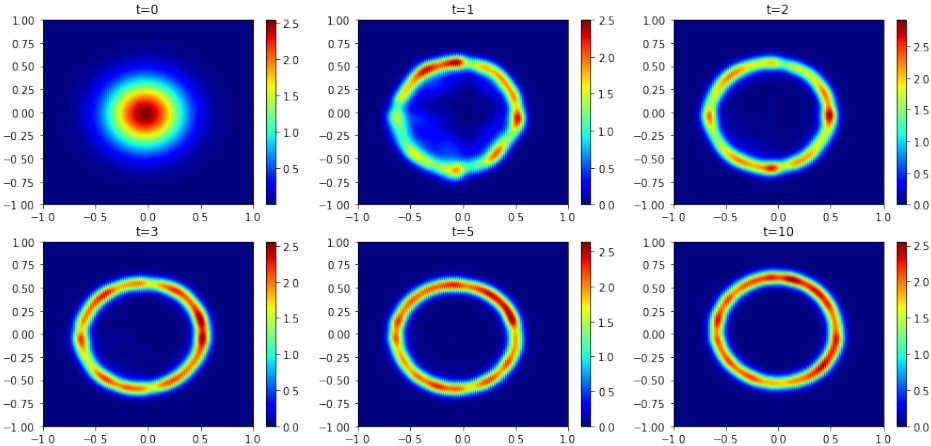

Figure 13: Density over time of the solution of the aggregation equation approximated with a FCNN.

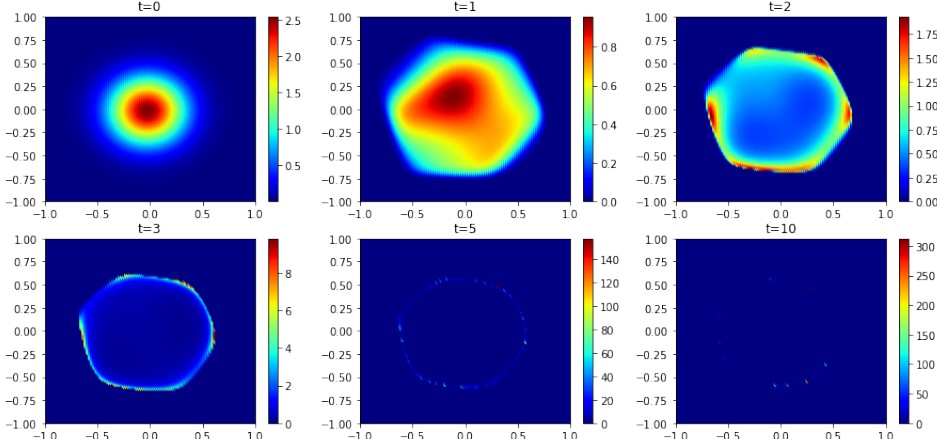

Figure 14: Density over time of the solution of the aggregation equation approximated with JKO-ICNN.

et al., 2013)), we show the densities over time obtained on a discrete grid (Figure 11), by learning the position of the particles (Figure 12) and with the FCNN (Figure 13). For the last two, the density reported is obtained with kernel density estimation where we chose by hand the bandwidth to match well the sampled points. We also report the evolution of the density for the JKO-ICNN scheme. In Figure 14, we show the evolution of the density obtained by the change of variable formula. We observe that values seem to explode which may due to the fact that the stationary solution may not have a density or to numerical unstabilities. We report in Figure 15 the densities obtained with a kernel density estimation. We also report the particles generated through a FCNN, the particle evolution and JKO-ICNN in Figure 16. We observe that the particles match perfectly the ring. For the FCNN, there seem to be some noise. JKO-ICNN recover also well the ring but particles seem to not be uniformly distributed over the ring.

For the SW-JKO scheme, we take $\tau = 0.05$ and run it for 200 steps (from $t = 0$ to $t = 10$) starting from $\mu_0 = \mathcal{N}(0, I)$. For JKO-ICNN, we choose $\tau = 0.1$ and run it for 100 steps as the diffusion was really long.

We take a grid of $50 \times 50$ samples on $[-1, 1]^2$. To optimize the weights, we used an SGD optimizer with a momentum of 0.9 with a learning rate of $10^{-4}$ for 300 epochs by inner optimization scheme. For particles, we optimized 1000 particles (sampled initially from $\mu_0$) with an SGD optimizer with a momentum of 0.9, a learning rate of 1 and 500 epochs by JKO step. We take a FCNN composed of 5 hidden layers with 64 units and leaky relu activation functions. They are optimized with an Adam optimizer with a learning rate of $10^{-4}$ (except for the first iteration where we take a learning rate of $5 \cdot 10^{-3}$) for 400 epochs by JKO step. For

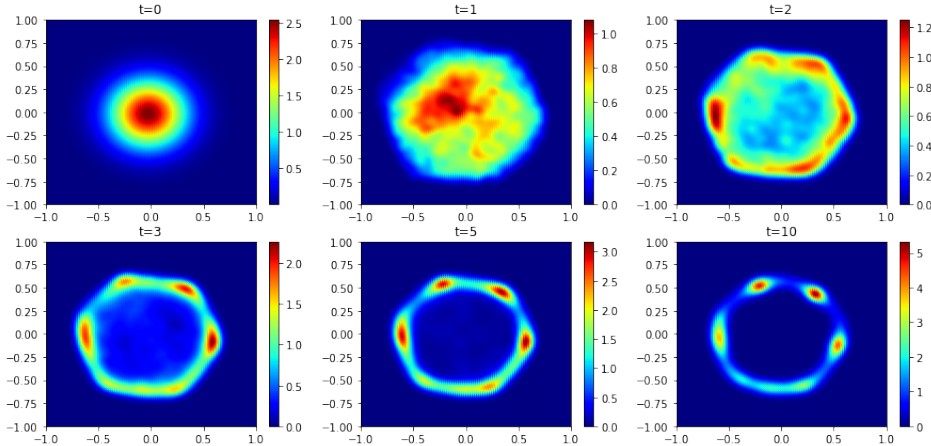

Figure 15: Density over time of the solution of the aggregation equation approximated with JKO-ICNN and kernel density estimator.

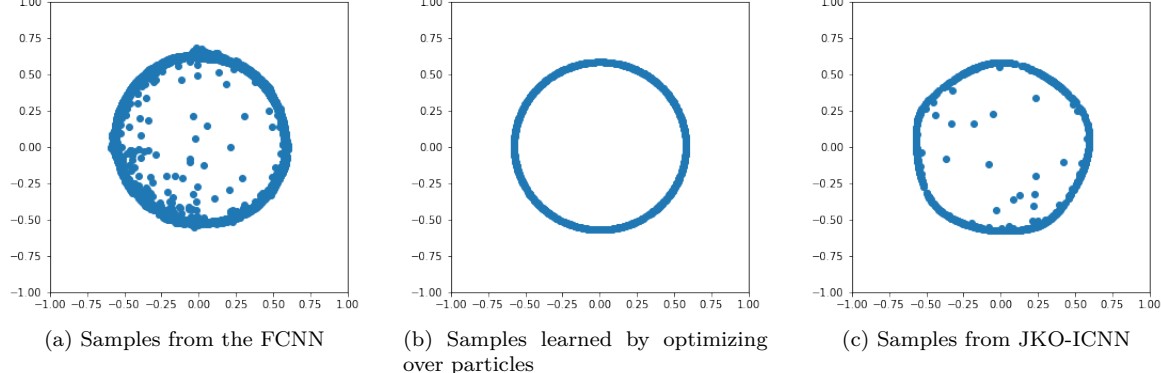

(a) Samples from the FCNN

(b) Samples learned by optimizing over particles

(c) Samples from JKO-ICNN

Figure 16: Samples of the stationary distribution

JKO-ICNN, we choose the same parameters as for the previous experiments. In any case, we take $n_\theta = 10^4$ projections to approximate $SW_2$ and a batch size of 1000.

On Figure 17, we plot on the two first columns the stationary density learned by the different methods. On the third column, we plot the evolution of the functional along the flows.

The functionals seem to converge towards the same value. However, JKO-ICNN seems to not be able to capture the right distribution. We also note that the curve for the FCNN is not very smooth, which can probably be explained by the fact that we take independent samples at each step.

On Figure 18, we show the evolution of the functionals along different learned flows. We observe that for the discretized grid, we obtain the worse results which is understandable since we have discretization error. The particle is the most stable as particle's position do not move anymore once the stationary state is reached. For the FCNN, we observe oscillations which are due to the fact that we take independent samples at each time t. Finally, the JKO-ICNN scheme seems to converge toward the same value as the SW-JKO scheme with FCNNs.

We also tried to use normalizing flows with this functional. We observed that the training seems harder than with the FCNN. Indeed, with simple flows such as RealNVP, the model has a lot of troubles of learning the Dirac ring, probably because of the hole. More generally, since normalizing flows are bijective transformations, they must preserve topological properties, and therefore do not perform well when the

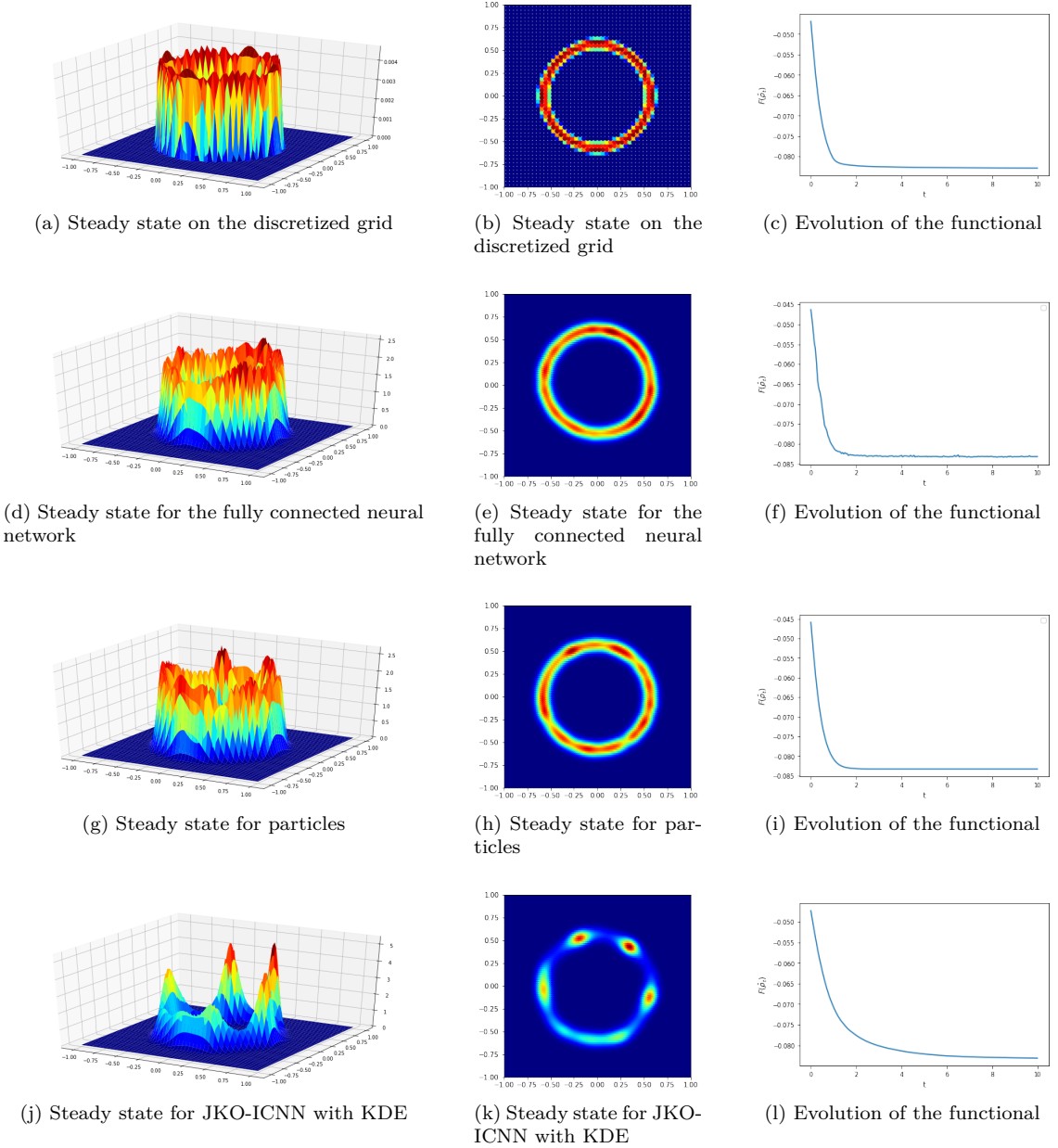

(a) Steady state on the discretized grid

(b) Steady state on the discretized grid

(c) Evolution of the functional

(d) Steady state for the fully connected neural network

(e) Steady state for the fully connected neural network

(f) Evolution of the functional

(g) Steady state for particles

(h) Steady state for particles

(i) Evolution of the functional

(j) Steady state for JKO-ICNN with KDE

(k) Steady state for JKO-ICNN with KDE

(l) Evolution of the functional

Figure 17: Steady state and evolution of the functional of the aggregation equation for $a = 4$, $b = 2$.

standard distribution and the target distribution do not share the same topology (*e.g.* do not have the same number of connected components or "holes" as it is explained in (Cornish et al., 2020)). For CPF, it worked slightly better, but was not able to fully recover the ring (at least at time $t = 5$) as we can see on Figure 19. Morover, the training time for CPF was really huge compared to the FCNN.

**Other functionals.** As in section 3 of Carrillo et al. (2021), we also tried to use

$$W(x) = \frac{\|x\|^2}{2} - \log(\|x\|) \tag{77}$$

as interaction potential with a FCNN. We find well that the steady state is an indicator function on the centered disk of radius 1 (Figure 20).

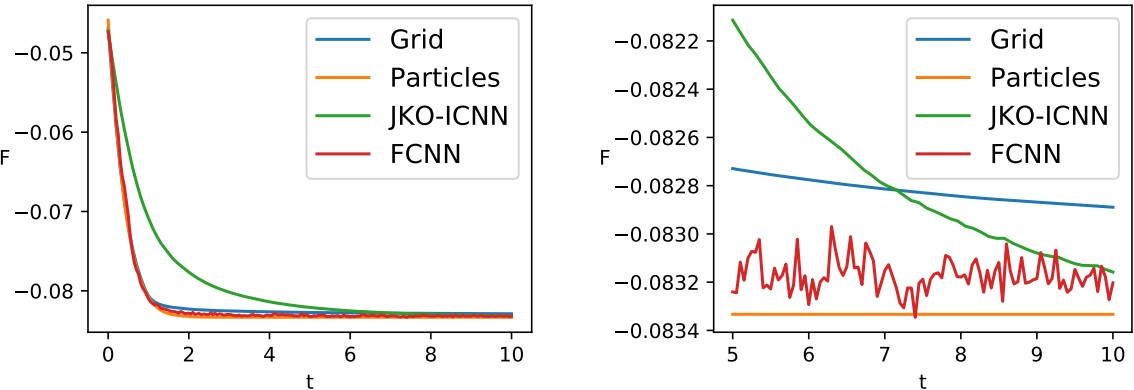

Figure 18: Evolution of the aggregation functional along different flows.

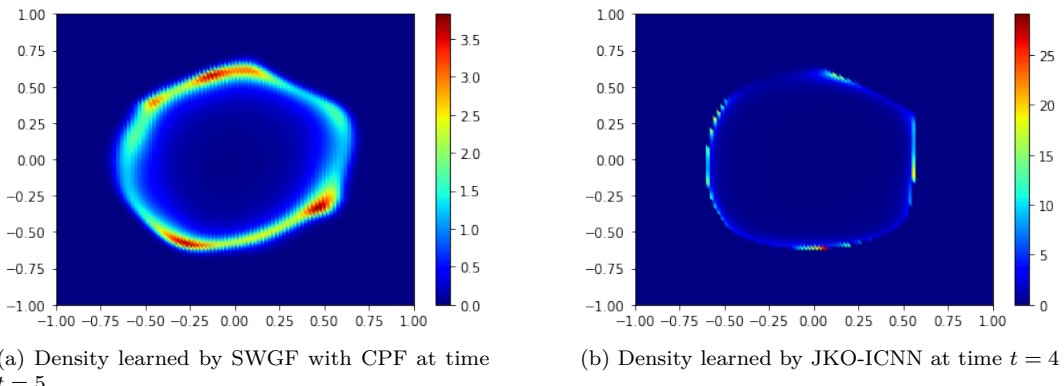

(a) Density learned by SWGF with CPF at time $t = 5$

(b) Density learned by JKO-ICNN at time $t = 4$

Figure 19: Density learned for the aggregation equation for CPF and JKO-ICNN.

Another possible functional without using internal energies is to add a drift term

$$\int V(x)\rho(x)\mathrm{d}x. \tag{78}$$

Then, the Wasserstein gradient flow is solution to

$$\partial_t \rho_t = \mathrm{div}(\rho\nabla(W * \rho)) + \mathrm{div}(\rho\nabla V). \tag{79}$$

Carrillo et al. (2021) use $W(x) = \frac{\|x\|^2}{2} - \log(\|x\|)$ and $V(x) = -\frac{\alpha}{\beta}\log(\|x\|)$ with $\alpha = 1$ and $\beta = 4$. Then, it can be shown (see Carrillo et al. (2021); Chen & Kolokolnikov (2014); Carrillo et al. (2015)) that the steady state is an indicator function on a torus of inner radius $R_\mathrm{i} = \sqrt{\frac{\alpha}{\beta}}$ and outer radius $R_\mathrm{o} = \sqrt{\frac{\alpha}{\beta} + 1}$ which we observe on Figure 21 and Figure 22.

For this last experiment, we observed some unstabily issues in the training phase. It may be due to non-locality of the interaction potential $W$ as it is stated in (Carrillo et al., 2021).

### D.5 Sliced-Wasserstein flows

In this experiment, we aim at minimizing the following functional:

$$\mathcal{F}(\mu) = \frac{1}{2}SW_2^2(\mu, \nu) + \lambda\mathcal{H}(\mu) \tag{80}$$

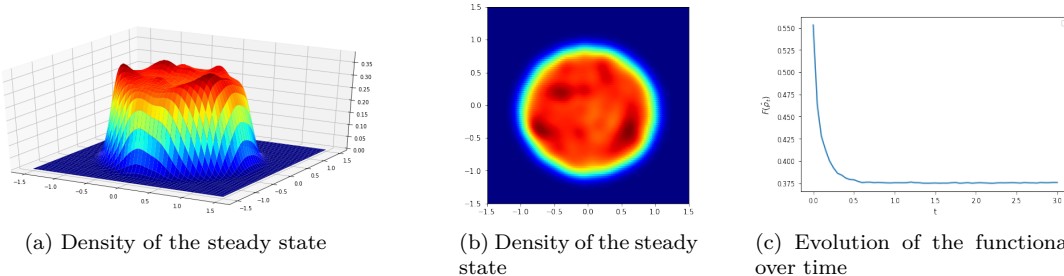

(a) Density of the steady state     (b) Density of the steady state     (c) Evolution of the functional over time

Figure 20: Steady state and evolution of the functional for $W(x) = \frac{\|x\|^2}{2} - \log(\|x\|)$.

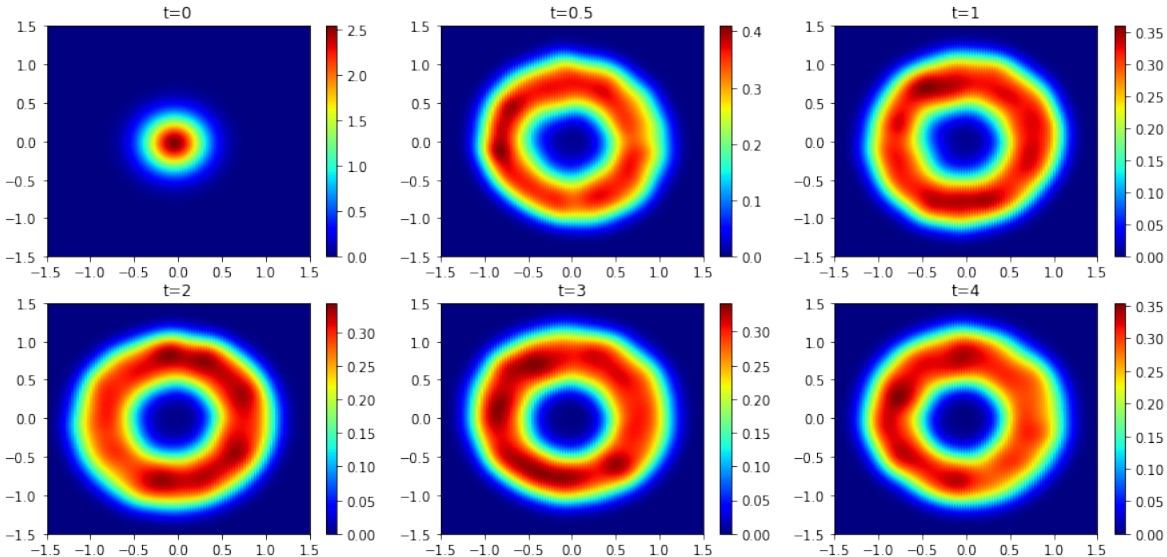

Figure 21: Density over time of the solution of the aggregation-drift equation approximated with a FCNN.

where $\nu$ is some target distribution from which we have access to samples. In this section, we use MNIST (LeCun & Cortes, 2010), FashionMNIST (Xiao et al., 2017) and CelebA (Liu et al., 2015).

We report the Fréchet Inception Distance (FID) (Heusel et al., 2017) for MNIST and FashionMNIST between $10^4$ test samples and $10^4$ generated samples. As they are gray images, we duplicate the gray levels into 3 channels. Moreover, we use the code of Dai & Seljak (2021) (available at https://github.com/biweidai/SINF) and reported their result for SWF (in the ambient space) and SWAE in Table 2. For SWF in the latent space, we used our own implementation.

**With a pretrained autoencoder.** First, we optimize the functional in the latent space of a pretrained autoencoder (AE). We choose the same AE as Liutkus et al. (2019) which is available at https://github.com/aliutkus/swf/blob/master/code/networks/autoencoder.py. We report results for a latent space of dimension $d = 48$.

In this latent space, we applied a FCNN and a RealNVP. In either case, we used $n_\theta = 10^3$ projections to approximate $SW_2$. We chose a batch size of 128 samples. The FCNN was chosen with 5 hidden layers of 512 units with leaky relu activation function. The RealNVP was composed of 5 coupling layers, with FCNN as scaling and shifting networks (with 5 layers and 100 hidden units). We trained the networks at each inner optimization step with a learning rate of $5 \cdot 10^{-3}$ for RealNVP (and $10^{-2}$ for the first iteration) and of $10^{-3}$ for the FCNN (and $5 \cdot 10^{-3}$ for the first itration) during 1000 epochs.

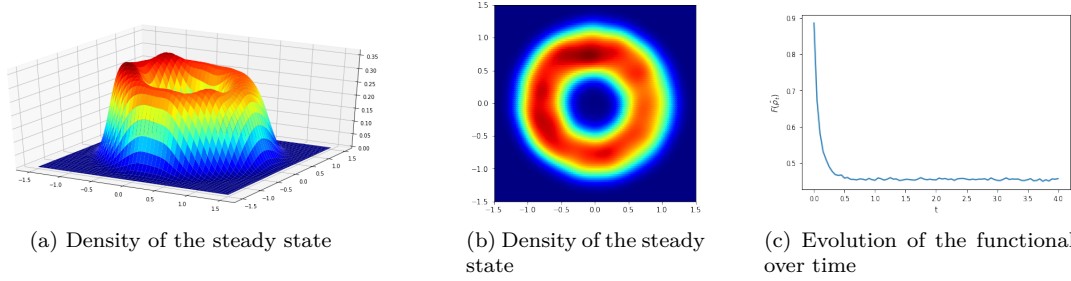

(a) Density of the steady state

(b) Density of the steady state

(c) Evolution of the functional over time

Figure 22: Steady state and evolution of the functional for the aggregation-drift equation.

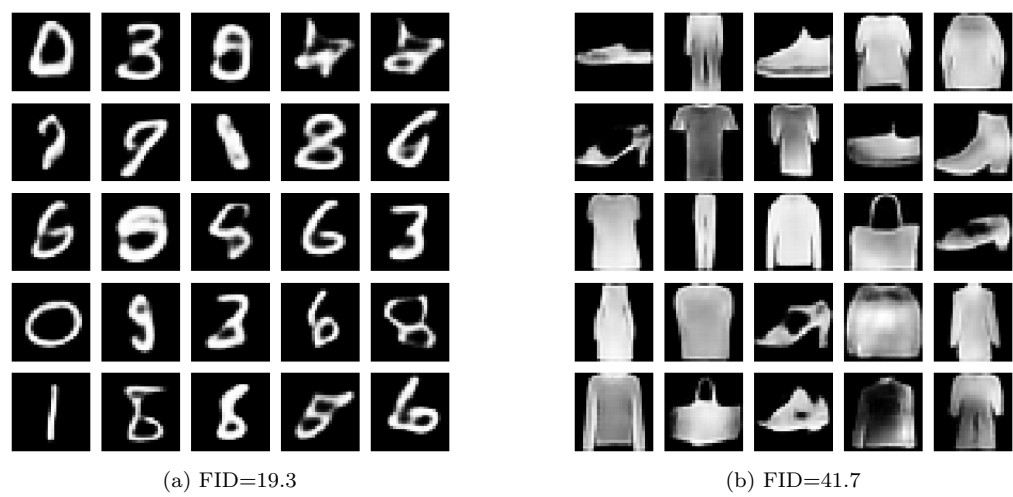

(a) FID=19.3

(b) FID=41.7

Figure 23: Generated sample obtained through a pretrained decoder + FCNN.

Following Liutkus et al. (2019), we choose $\tau = 0.5$ and $\lambda = 0$. We run it for 10 outer iterations.

We report in Figure 23 the result obtained with FCNN on MNIST and FashionMNIST. Overall, the results seem quite comparable and our method seems to perform well compared to other methods applied in latent spaces.

On Figure 24, we report samples obtained with RealNVP on MNIST, FashionMNIST and CelebA. For CelebA, we choose the same autoecoder with a latent space of dimension 48 and we ran the SW-JKO scheme for 20 steps with $\tau = 0.1$.

We can also optimize directly particles in the latent space and we show it on CelebA on Figure 25 for $\tau = 0.1$ and for 10 steps.

**In the Original Space.** We report here the results obtained by running the SW-JKO scheme in the original spaces of images, which are very high dimensional ($d = 784$ for MNIST and FashionMNIST). We obtained worse results than in the latent space. Notice that we used only 1000 projections, and ran it for 50 outer iterations with a step size of $\tau = 5$.

On Figure 26, we use a RealNVP and add a uniform dequantization (Ho et al., 2019) and learned it in the logit space as it is done in (Dinh et al., 2016; Papamakarios et al., 2017) because using normalizing flows need continuous data. The RealNVP is composed here of 2 coupling layers with FCNN also composed of 2 layers and 512 hidden units. We choose $\tau = 5$, and run it for 20 steps. At each inner optimization problem, the neural networks are optimized with an Adam optimizer with a learning rate of $10^{-2}$ and for 500 epochs.

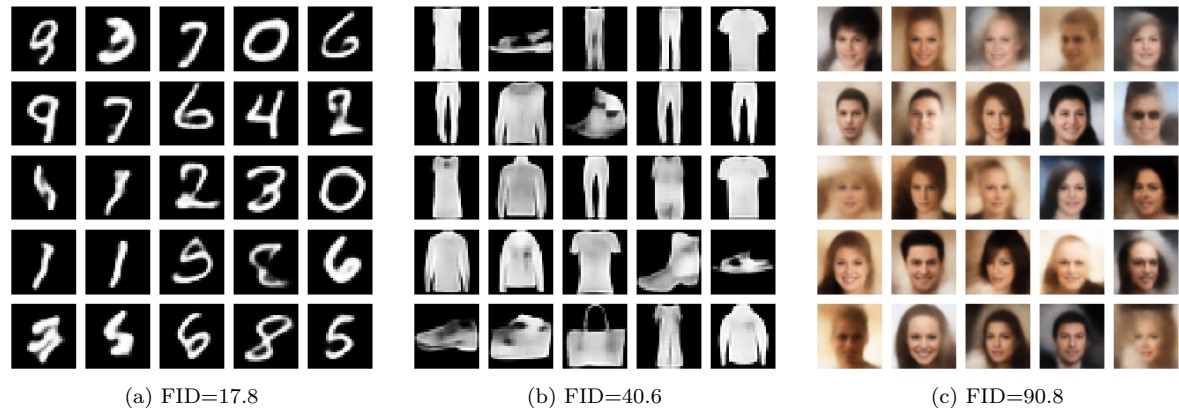

(a) FID=17.8 (b) FID=40.6 (c) FID=90.8

Figure 24: Generated sample obtained through a pretrained decoder + RealNVP.

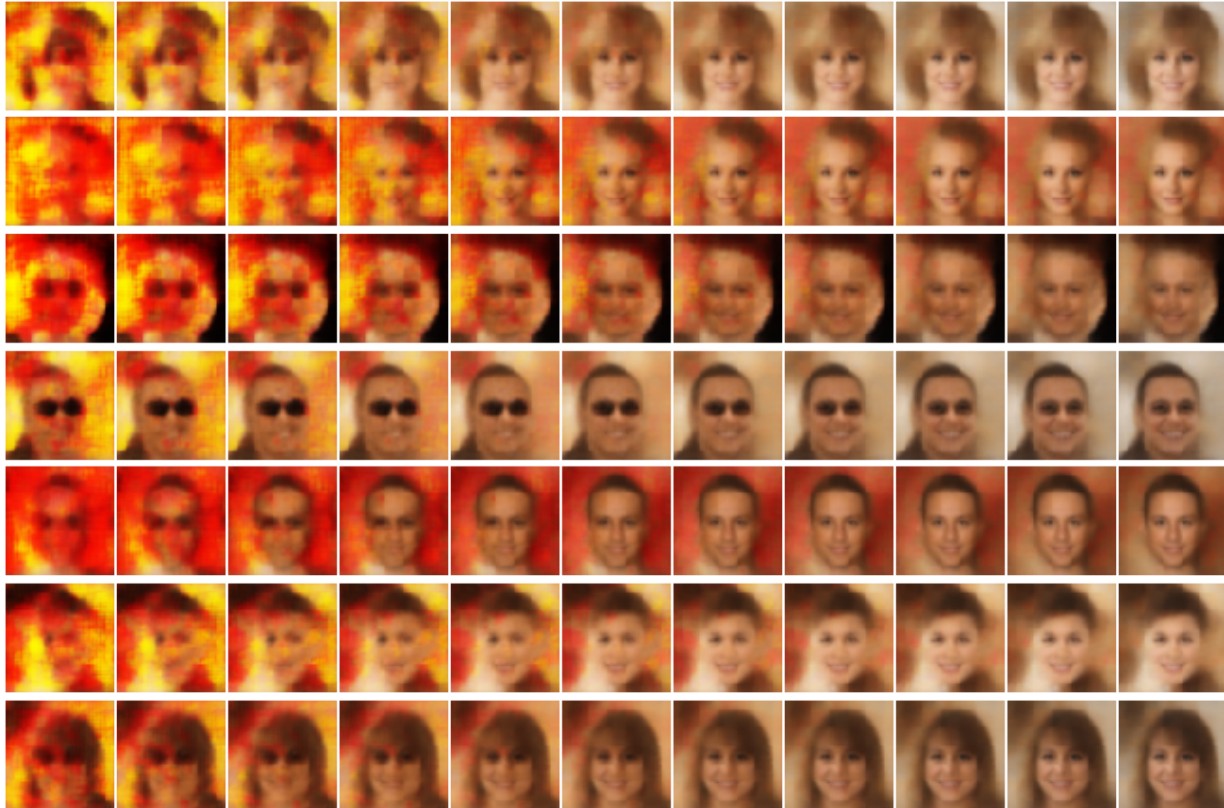

Figure 25: Particle through SW-JKO with $\tau = 0.1$ for 10 steps.

We also report results obtained using a convolutional neural network (CNN) in Figure 27. The idea here is that we can capture inductive bias, as it is well known that CNNs are efficient for image-related tasks. We obtained a slightly better FID on MNIST, but worse results on FashionMNIST. In term of quality of image, the generated samples do not seem better.

For the CNN, we choose a latent space of dimension 100, and we first apply a linear layer into a size of $128 \times 7 \times 7$. Then we apply 3 convolutions layers of (kernel_size, stride, padding) being respectively $(4, 2, 1)$, $(4, 2, 1)$, $(3, 1, 1)$. All layers are followed by a leaky ReLU activation, and a sigmoid is applied on the output.

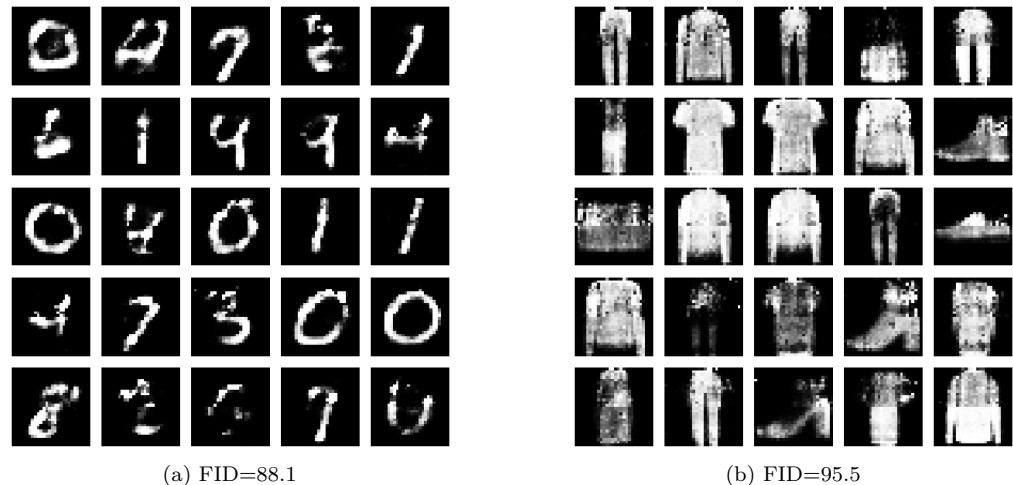

(a) FID=88.1

(b) FID=95.5

Figure 26: Generated sample obtained in the original space with RealNVP.

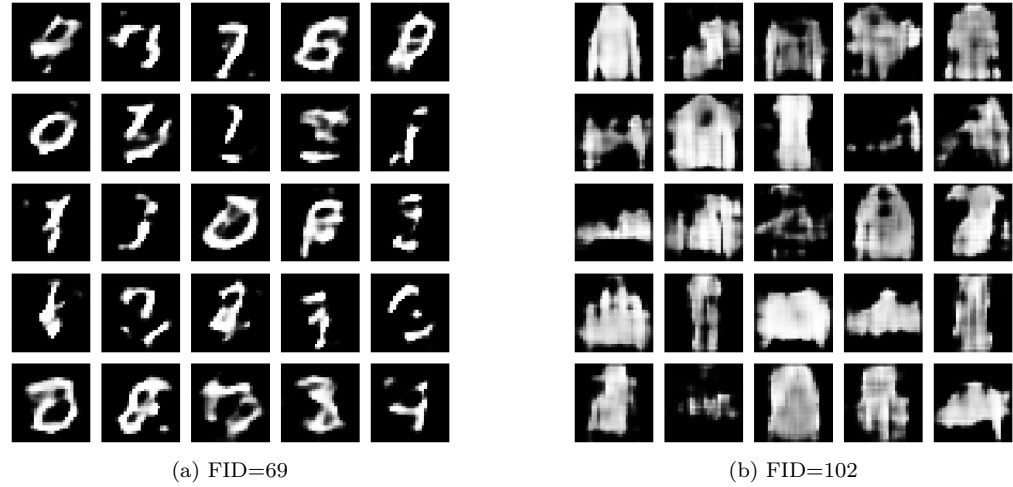

(a) FID=69

(b) FID=102

Figure 27: Generated sample obtained in the original space with CNN.

Nevertheless, samples obtained in the space of image look better than those obtained through the particle scheme induced in (Liutkus et al., 2019).

