# OpenReview forum: "Efficient Gradient Flows in Sliced-Wasserstein Space"
_TMLR — Accepted by TMLR_

### Review · Reviewer_xUbb · 2022-07-26

**Summary Of Contributions:**

The paper is concerned with gradient flows on the space of probability distributions, where sliced Wasserstein distance is used as a metric. To define and implement the gradient flow, the JKO procedure is employed where exact Wasserstein distance is replaced with its sliced version. Although there is no explicit relationship between the two gradient flows, the paper provide some insight into this by studying certain examples, supported by numerical evidence. In order to numerically implement the flow, the paper proposes to represent the probability distribution at each step as either, a discrete distribution on fixed set of points, or push-forward of a generator map $ g_\theta \sharp p_z$ where $p_z$ is a Gaussian distribution in the latent space. At each JKO step, the discrete distribution or the parameters of the generator are updated to minimize the JKO objective. The proposed algorithm is studied and compared with the related approaches with the aid of several numerical examples involving synthetic and real data. The functionals used in the paper are: relative entropy with respect to a target distribution, or the interaction energy.

**Requested Changes:**

Comments:

My main concern is about the motivation for using SW and its advantages. According to my understating, the main bottleneck in implementing these type of gradient flows is computation of the functional in terms of particles, not approximating the Wasserstein distance. Using ICNN $phi$, the W2 distance between p and $\nabla \phi \sharp p$ is simply $\int |\nabla \phi(x)-x|^2 dp$. In case a map $T(x)$ is used instead of $\nabla \phi$, one can use the $\int |T(x)-x|^2 dp$ which for optimal $T$ becomes the W2 distance. However, computing the functional, specially entropy is not straightforward. The existing approaches either used the approximation suggested in the paper in (24), or its variational form which involves extra maximization step. Eq (24) involves $\log (\det (\nabla g_\theta(z)))$ which is expensive to compute and need to be approximated.

Putting the application of SW aside, another main difference of the proposed approach is using a single generator map from the latent space, instead of constructing sequence of maps as in the existing works. This is the reason for reduction from $K^2$ to $K$ compared to previous approaches and might be the reason for better performance in numerical experiments.

In summary, I understand the computational value of SW compared to W2 in general when two distributions are compared, but I do not see its value in the gradient flow context.

Questions/Changes:
- Is it possible to implement your proposed approach, but simply use $\int |g_\theta^k(z) - g_\theta^{k+1}(z)|^2 p_z dz$ as regularizer instead of SW? This might be a good proxy for W2.

- Is it possible to derive the limiting PDE? I see you derive the stationary conditions for JKO step in the appendix.

- Are there any reasons for experiments reported in figure 2 to run till $d=12$?

- I do not understand the emphasis on SW being differentiable. Isn't W2 differentiable too?

- first line in background section: "bet" -> "be"


**Strengths And Weaknesses:**

Strength:
- nice introduction to the topic and proposed approach
- insight about the sliced Wasserstein flow compared with exact one
- broad numerical experiments and attention to approximation problems regarding SW

Weakness:
- the evidence and motivation for using SW was not strong
- no discussion on the existence of the gradient flow with SW, in the limit as JKO step-size converges to zero (what is the pde that describes this flow?)

---

> ### Author Response · Authors · 2022-08-19
> **Response to Reviewer xUbb (Part 1)**
>
> We thank the reviewer for their comment. Below, we address their questions and concerns.
>
> **The evidence and motivation for using SW was not strong.**
>
> In Section 3.1, we discuss different motivations to use SW in place of W. The first one is the computational properties of SW which is much cheaper to compute compared to the Wasserstein distance since SW has a $O(n\log n)$ complexity against $O(n^3\log n)$ complexity for W. Moreover, SW has a sample complexity which depends on the dimension only through the Monte-Carlo approximation. These properties make it interesting to use, but provide no clue on whether or not its gradient flows will have good properties.
>
> But the SW distance is also related to the Wasserstein through different properties which make it interesting to use as a proxy. The main one is that it shares the same topology as the Wasserstein distance [1] and the distances are equivalent. Moreover, the geodesics in SW space are connected with those in Wasserstein space [2], see Appendix A.1.
>
> These different properties provide many motivations to use SW as a proxy of W in order to compute gradient flows.
>
>
> **No discussion on the existence of the gradient flows with SW, in the limit as JKO step size converges to zero (what is the pde that describes this flow?)**
>
> In Section 3.2, we discuss the limit PDE of the SW gradient flows in some particular cases. Namely, when distributions are restricted to one dimensional subspaces, or when distributions are isotropic Gaussians, we can relate the SW-JKO scheme with the JKO scheme, and hence we obtain in the limit $\tau\to 0$ the same dynamics with a dilation parameter of $d$. Hence in these cases, the PDE is the same. Without the factor $d$, it adds a constant $d$ in the PDE. For example, in the Fokker-Planck case, we would obtain $\partial \rho_t = d (\mathrm{div}(\rho\nabla V) + \Delta \rho)$. We will add a discussion about it in the next version of the paper.
>
> For the more general case, it is still an open question whether or not the dynamic of Sliced-Wasserstein gradient flows is the same as Wasserstein gradient flow or not.
>
> **The main bottleneck in implementing these types of gradient flows is the computation of the functional in terms of particles, not approximating the Wasserstein distance.**
>
> While computing the functional term can be complicated when it involves the density, the computation of the Wasserstein distance is not easy.
>
> We agree that concomitant works proposed to use the Monge formulation to avoid to compute the Wasserstein distance, and in this case alleviate this part of the problem. But it adds some complexity to the problem. Indeed, we either need to use the gradient of a ICNN [3,4] to be sure to have a Monge map by Brenier’s theorem, and in this case, it adds a lot of computation cost as described in the paper. The other solution proposed in [5] and which was first released at the same time as this work, is to directly use the Monge formulation. In this case, the class of maps among which they minimize is huge and hence the optimization problem is more difficult (Note that in the works trying to approximate the Wasserstein distance, this formulation is rarely used, probably because of this issue).
>
> Moreover, as underlined by the reviewer, one bottleneck of using the Monge formulation in the JKO scheme is that it requires $O(K^2)$ evaluations of the networks, versus $O(K)$ when parameterizing the probability independently from previous probabilities.
>
> Note also that some works released in parallel of ours approximate directly the Wasserstein distance without using the Monge formulation (see e.g. [6]). And to do that, they rely on a method introduced in [7] which needs to approximate a potential and its conjugate using ICNNs. In contrast, with our method, we only require to compute SW which is very efficient to compute.
>
> For the functionals, we argue that the main difficulty is to evaluate the density (for example for the negative entropy). We propose here to use normalizing flows in order to solve this problem, when in [5], they propose a variational formulation and in [3, 4], they use the Monge map using that it is the gradient of a strictly convex function. Using normalizing flows has the advantage to alleviate the cost of evaluating the log likelihood.

---

> > ### Author Response · Authors · 2022-08-19
> > **Response to Reviewer xUbb (Part 2)**
> >
> > **Is it possible to implement your proposed approach, but simply use $\int |g_\theta^k(z)-g_\theta^{k+1}(z)|^2 p(z)\mathrm{d}z$ as regularizer instead of SW?**
> >
> > It is possible to use different proxies for the Wasserstein distance, as e.g. in [8,9]. SW has however the advantage of well defining a gradient flow on its own since it is a distance. Thus, even if we do not yet know its theoretical properties, the gradient flow is well defined, even if it is not seen as a proxy of Wasserstein.
> >
> > **Is it possible to derive the limiting PDE?** It is still an open question to derive what is the limiting PDE in general for sliced-Wasserstein gradient flows. As previously described, we know what it is in some particular cases by linking it with Wasserstein gradient flows.
> > In Appendix A.2, we derive (informally) the limiting PDE for Wasserstein gradient flows of the Fokker-Planck functional. But doing the same derivation for Sliced-Wasserstein gradient flows seems to be more difficult and would need to be investigated further, as the analogous of Equation (41) for the first variation of SW has not yet been proven.
> >
> > **Are there any reasons for experiments reported in Figure 2 to run till $d=12$?** This choice of dimension is arbitrary. However, it already shows the benefit of running SW gradient flows with normalizing flows compared to Wasserstein gradient flows with Euler-Maruyama or JKO scheme which has numerical instabilities. In Appendix D.2, in Figure 10, we reported the evolution for higher dimensions (until $d=100$).
> >
> > **I do not understand the emphasis on SW being differentiable. Isn't W2 differentiable too?**
> >
> > In practice, we can differentiate W2 with automatic differentiation in many libraries. Theoretically however, this is more complicated since the target measure is a constraint in the primal form (see e.g. Remark 4.5 of [10] or [11] for discussions about the differentiability of the Wasserstein distance, and the difficulty to evaluate its gradients ).  By using the dual form, some results can be derived but we only have access to a subgradient w.r.t. the marginal (see e.g. Proposition 1 in [12], Section 4.1).  We will add a more complete discussion about this in the next version of the paper (see the paragraph “Computational Properties”).
> >
> > **Typo.** Thank you for pointing this out. We will fix it in the revised version.
> >
> > [1] Nadjahi, Kimia, et al. "Asymptotic guarantees for learning generative models with the sliced-wasserstein distance." Advances in Neural Information Processing Systems 32 (2019).
> >
> > [2] Candau-Tilh, Jules. “Wasserstein and Sliced-Wasserstein Distances”. Master’s thesis, Université Pierre et Marie-Curie (2020).
> >
> > [3] Alvarez-Melis, David, Yair Schiff, and Youssef Mroueh. "Optimizing functionals on the space of probabilities with input convex neural networks." arXiv preprint arXiv:2106.00774 (2021).
> >
> > [4] Mokrov, Petr, et al. "Large-scale wasserstein gradient flows." Advances in Neural Information Processing Systems 34 (2021): 15243-15256.
> >
> > [5] Fan, Jiaojiao, Amirhossein Taghvaei, and Yongxin Chen. "Variational Wasserstein gradient flow." arXiv preprint arXiv:2112.02424 (2021).
> >
> > [6] Hwang, Hyung Ju, et al. "The deep minimizing movement scheme." arXiv preprint arXiv:2109.14851 (2021).
> >
> > [7] Korotin, Alexander, et al. "Wasserstein-2 generative networks." arXiv preprint arXiv:1909.13082 (2019).
> >
> > [8] Lin, Alex Tong, et al. "Wasserstein proximal of GANs." International Conference on Geometric Science of Information. Springer, Cham, 2021.
> >
> > [9] Cancès, Clément, Thomas O. Gallouët, and Gabriele Todeschi. "A variational finite volume scheme for Wasserstein gradient flows." Numerische Mathematik 146.3 (2020): 437-480.
> >
> > [10] Peyré, Gabriel, and Marco Cuturi. "Computational optimal transport: With applications to data science." Foundations and Trends® in Machine Learning 11.5-6 (2019): 355-607.
> >
> > [11] Genevay, Aude, Gabriel Peyré, and Marco Cuturi. "Learning generative models with sinkhorn divergences." International Conference on Artificial Intelligence and Statistics. PMLR, 2018.
> >
> > [12] Cuturi, Marco, and Arnaud Doucet. "Fast computation of Wasserstein barycenters." International conference on machine learning. PMLR, 2014.

---

### Review · Reviewer_nXjA · 2022-08-03

**Summary Of Contributions:**

The paper introduces the concept of sliced Wasserstein gradient flows and employs them to minimize the functionals on the space of the probability measures. The minimization of functionals is done according to the sliced wasserstein JKO stepping scheme (SW-JKO) which the authors propose.

The SW-JKO scheme is a natural modification of the original JKO scheme (W distance is simply replaced by SW distance in the JKO stepping). JKO is a procedure (introduced by Jordan-Kinderlehrer-Otto) to compute the (conventional) Wasserstein-2 gradient flows of functionals of probability measures. The latter scheme can be viewed as a time-discretized gradient descent in the spaces of probability measures. While the scheme works in theory, its practical implementation is challenging as each JKO step is itself a non-trivial optimization over probability measures involving the Wasserstein-2 distance term which is hard to estimate. There exist several works establishing numerical precedures for to handle this issue (the authors discuss them in Section 2.3).

In this paper, the authors propose a numerical procedure based either on using a generative model, particle-based approximation or grid discretization to implement SW-JKO stepping in practice. According to the provided computational results, SW-JKO converges to the minimum of the functionals (sometimes) faster (or better) than the alternative methods which compute (conventional) wasserstein gradient flows such as the JKO-ICNN methods (appearing in several related papers), etc.

**Broader Impact Concerns:**

My main concern about this paper is that the concept introduced in the paper – SW gradient flows (and the related SW JKO) – seems to be unnatural and presumably will have limited impact on the machine learning community/practitioners. Let me explain my concern below.

First, when we talk about the Wasserstein gradient flows, there are two (related) tasks in view:
1) Computing the gradient flow dynamics (trajectories);
2) Computing the “end” of the gradient flow (a minimum of the functional).

The gradient flow (1) itself seems to be an important tool to analyse the **dynamics** of stochastic processes appearing in a wide range of sciences and their practical applications (especially those fields where the Fokker-Plank equation appears). To the best of my understanding, the prior related works (Fan et al., 2021; Alvarez et al., 2021; Mokrov et. al. 2021) propose algorithms which are capable of recovering the entire dynamics of the flow. As far as I understand, the current work is unable to do so as it recovers Sliced Wasserstein dynamics, which differs from the Wasserrstein gradient flow (the relation is discussen in many places in the paper). Thus, the applicability of the method proposed here to studying dynamics appearing in real-world problems, e.g., those characterized by FPE, seems questionable.

The current paper focuses only on aspect (2), namely **minimization of the functionals** (computing the end of the flow). This immediately raises the question: why do we need to optimize the functional with SW-JKO (or any other JKO scheme). Why not to just directly pick a generative model (invertible in the case when the functional contains entropy) and directly optimize its parameters to minimize the functional? Or not to pick a set of particles and directly optimize them with the gradient descent? Why is SW-JKO better? Unfortunately, the current paper does not provide a deep discussion/evaluation related to this important question. This makes not very clear whether SW-JKO indeed provides any improvement in optimizing functionals.

To conclude, in my view, to show the potential superiority of SW-JKO to optimize functionals, it is essential to conduct more detailed experimental evaluation and comparisons, see the requested changes section. Unfortunately, otherwise SW-JKO seems to be just a nice mathematical concept but without potential applications in practice/extensions

Nevertheless, since the acceptance criteria for TMLR is technical correctness and clarity of presentation (which are ok in this paper) rather than significance or impact (which are, in my opinion, questionable), I recommend to push this paper forward to the next revision.

**Requested Changes:**

Experiments

The current work mostly considers the problem of optimizing functionals over probability distributions. The authors claim that their methods performs (sometimes) faster or better than existing alternatives to minimize the functionals. However, the experimental evidence of this is limited. Thus, I suppose the following experiments should be conducted in the revision:

In the setting with **neural networks/generative models** approximation. I would expect here both quantitative and qualitative comparisons, e.g., it would be nice to see some non-trivial multi-modal stationary distributions here (related works do consider such setups).
1) Comparison with (Fan et. al, 2022) [ICML 2022] paper. In the current paper, I see comparisons only with JKO-ICNN (Mokrov et. al., 2021), but (Fan et. al., 2022) claim to outperform JKO-ICNN by a noticeable margin, see their Figure 5.4. Besides, I wonder why the authors do not compare with JKO-ICNN (and Fan et al.) in recovering non-Gaussian the stationary distributions. For examples, why does Figure 9 contain JKO-ICNN but figure 8 - does not (explanation in Fig. 2b is not very convincing)?
2) Comparison with the direct optimization baselines. Namely, one may pick a generative model (or a normalizing flow) and directly optimize the functional with it (optimize its parameters with SGD).

Comparisons in the **particle-based** setting. In Appendices, the authors formulate their particle-based SW-JKO approach. Thus, it seems reasonable to request to compare it with the direct particle-based optimization of some functionals. This is neede to better understand how does the proposed scheme works compared to the straightfoward solution.

Same applies to the **grid-based** optimization SW-JKO presented in appendices.

In the above metnioned experiments, could the authors please report both final metrics & computational times?

Missing references

Jean-David Benamou, Guillaume Carlier, Quentin Mérigot, and Edouard Oudet. Discretization of functionals involving the Monge–Ampère operator. Numerische mathematik, 134(3):611–636, 2016.

**Strengths And Weaknesses:**

Strength
1) The method works better/faster than other related methods in some experimental scenarios (JKO-ICNN method in the bayesian logistic regression (table 1), SWF method in generative modeling (Figure 5, Table 2)).
2) The method is more flexible than JKO-ICNN since SW enjoys a closed form differentiable approximation.
3) The writing of the paper is clear
4) Technically the paper seems to be correct with no major flaws

Weaknesses
1) Broader impact/significance seems to be limited (see the Broader Impact section);
2) In general, the experimental evidence is insufficient to support some claims;
3) Some important comparisons to related works are missing;
4) The (straightforward) direct optimization baseline seems to be missing;

---

> ### Author Response · Authors · 2022-08-19
> **Response to Reviewer nXjA (Part 1)**
>
> We thank the reviewer for their comments. Below, we address its questions and concerns.
>
>
> **Non-trivial multi-modal stationary distribution.**
>
> We experiment with simple setups on $\mathbb{R}^2$ where we have access to closed forms (i.e. with Gaussians). We believe that experimenting on e.g. mixture of Gaussians would not bring more since it would mainly rely on the expressivity of the normalizing flow more than on SW-JKO. Moreover, we conducted non-trivial experiments with neural networks for the aggregation equation (see Figure 4, Section  4.2 and Appendix D.4). We also conducted experiments on bayesian logistic regression (see Section 4.1) which are not trivial.
>
> We refer to the general response for a more thorough discussion about targeting multi-modal stationary solutions.
>
> **Comparison with [1].** Our work first appeared on arXiv (whose link we do not provide for anonymity) at the same time of [1] when the main competitor for JKO scheme was JKO-ICNN, and their code appeared very recently on Github. Hence, we did not compare ourselves with their work, but they are definitely a strong competitor. However, we would like to underline that their contribution is mainly to tackle functionals which use the density and which have a variational formulation. Moreover, it involves solving a minimax problem which can be hard to optimize in practice. Hence, their contribution could be seen as complementary as ours since we could use their method within the SW-JKO scheme.
>
> **I wonder why the authors do not compare with JKO-ICNN (and Fan et al.) in recovering non-Gaussian the stationary distributions. For examples, why does Figure 9 contain JKO-ICNN but figure 8 - does not (explanation in Fig. 2b is not very convincing)?**
>
> To recover the non Gaussian stationarity (Figure 2), we did not compare with JKO-ICNN since we experienced many numerical issues, We reported on Figure 2.b the numerical issues observed in dimension 10. Note that [1] also reported experiencing numerical issues with JKO-ICNN.
>
> Figure 9 does contain JKO-ICNN since we compare here the results with the groundtruth of Wasserstein gradient flows at time 0.5 and 0.9. Note that we do not know yet if Sliced-Wasserstein gradient flows are similar to Wasserstein gradient flows. We do not report it on Figure 7 and 8, since we were only interested in the evolution of the mean and variance of the SW-JKO scheme to see if they follow the same dynamic as Wasserstein gradient flows (which we know in this case in closed-form). For JKO-ICNN, it is known since it is an approximation of Wasserstein gradient flows and it has been experimented for different functionals in [3].
>
> **Comparison with the direct optimization baselines. Namely, one may pick a generative model (or a normalizing flow) and directly optimize the functional with it (optimize its parameters with SGD).**
>
> We agree that we may directly optimize the functional. Note that the same problem occurs in [1,2,3]. Moreover, we note that it is hard to do a fair comparison, since we do not solve the same problem. Indeed, it is not clear how to choose the number of epochs while having a fair comparison between the two.
>
> Finally, we experimented with comparing between directly optimizing the reverse KL (the usual loss of normalizing flows) and the SW-JKO scheme (see paragraph Gaussian case in Section 4.1) and we noted that we obtain similar results in general. However, when augmenting the dimension, we observed a lot more instabilities by directly optimizing the reverse KL compared to the SW-JKO scheme. Hence, we argue that adding the SW distance acts as a regularizer which has a benefit in terms of training stability.
>
> We also would like to note that directly minimizing the functional using SGD is a gradient descent on parameters, but not a gradient flow in the space of probability measure. Hence, the two methods are not really comparable. However, we acknowledge that, at the end of the day, since we need to approximate the probability measures using neural networks, we need to perform gradient descent in the space of parameters.
>
> We refer to the general response for more insights about the comparison with directly minimizing the functional.
>
> **Missing references.** We thank the reviewer and will add this missing reference in the revised version.

---

> > ### Author Response · Authors · 2022-08-19
> > **Response to Reviewer nXjA (Part 2)**
> >
> > **The gradient flow (1) itself seems to be an important tool to analyse the dynamics of stochastic processes appearing in a wide range of sciences and their practical applications (especially those fields where the Fokker-Plank equation appears). To the best of my understanding, the prior related works (Fan et al., 2021; Alvarez et al., 2021; Mokrov et. al. 2021) propose algorithms which are capable of recovering the entire dynamics of the flow.**
> >
> > First, we would like to note that even though [1,2,3] can indeed recover the whole dynamics, they are mostly interested in the final solution, just like us. Moreover, we can also recover the whole dynamics but directly in terms of measures, we cannot indeed in general follow an individual particle.
> >
> > Note also some other works (first released after ours on arXiv) which use the same type of formulation (push forward of standard noise) with the Wasserstein distance [4]. In this case, they need to compute the Wasserstein distance by using different methods such as [5] which are computationally heavy.
> >
> > **As far as I understand, the current work is unable to do so as it recovers Sliced Wasserstein dynamics, which differs from the Wasserstein gradient flow (the relation is discussed in many places in the paper).**
> >
> > We recover the Sliced-Wasserstein dynamics which are currently theoretically unknown. However, we showed in this work many empirical results showing that the dynamics are very similar to the Wasserstein gradient flows dynamics (see e.g. Figure 1).
> >
> > Here, we hope to pave the way toward theoretical study of these flows and hopefully show that they follow the same dynamics.
> >
> > **Why do we need to optimize the functional with SW-JKO (or any other JKO scheme).**
> >
> > We agree that it is an important question which is not really answered in the concurrent works [1,2,3]. Here, we experimented more training stability by comparing with optimizing directly the reverse KL to train the normalizing flow and by regularizing with SW.
> >
> > However, note that JKO scheme provides better theoretical results ensuring the existence of a unique minimizer… Here, for SW-JKO, we do not have yet such results since the SW space is not a geodesic space and hence the theory introduced in [6] cannot be used and the theoretical results are much harder to prove than in the case of the Wasserstein distance. But we hope to pave the way towards such study.
> >
> > Moreover, while we agree that in machine learning, the main interest is often on the minimum of the functional (and this is the main point of view that we took in our work), some works such as [7] considered the JKO scheme to study the underlying dynamic of trajectories (by learning the functional at each time step instead of learning).
> >
> > We refer to the general discussion for more insights about the benefit of using a JKO scheme versus directly minimizing a functional.
> >
> > **SW-JKO seems to be just a nice mathematical concept but without potential applications in practice/extensions.**
> >
> > We respectfully disagree with this sentence since we are able to minimize similar functionals as in [1,2,3]  with much less computational complexity. Moreover, we believe that the theoretical questions raised in this paper on the dynamics of Sliced-Wasserstein gradient flows is of much interest, and we hope that it will raise interest to solve this problem.
> >
> >
> > [1] Fan, Jiaojiao, Amirhossein Taghvaei, and Yongxin Chen. "Variational Wasserstein gradient flow." arXiv preprint arXiv:2112.02424 (2021).
> >
> > [2] Mokrov, Petr, et al. "Large-scale wasserstein gradient flows." Advances in Neural Information Processing Systems 34 (2021): 15243-15256.
> >
> > [3] Alvarez-Melis, David, Yair Schiff, and Youssef Mroueh. "Optimizing functionals on the space of probabilities with input convex neural networks." arXiv preprint arXiv:2106.00774 (2021).
> >
> > [4] Hwang, Hyung Ju, et al. "The deep minimizing movement scheme." arXiv preprint arXiv:2109.14851 (2021).
> >
> > [5] Korotin, Alexander, et al. "Wasserstein-2 generative networks." arXiv preprint arXiv:1909.13082 (2019).
> >
> > [6] Ambrosio, Luigi, Nicola Gigli, and Giuseppe Savaré. Gradient flows: in metric spaces and in the space of probability measures. Springer Science & Business Media, 2005.
> >
> > [7] Bunne, Charlotte, et al. "Proximal Optimal Transport Modeling of Population Dynamics." International Conference on Artificial Intelligence and Statistics. PMLR, 2022.

---

### Review · Reviewer_zKX6 · 2022-08-05

**Summary Of Contributions:**

Consider minimising a function $\mathcal{F}(\mu)$ on the space $\mu \in P_2$ of probability measures with finite second moment. One can use a proximal-gradient type method and set $$\mu_{k+1} = \text{argmin}_{\mu \in P_2} \quad J(\mu) + \frac{1}{2 \tau}\text{dist}^2(\mu, \mu_k)$$ for some appropriate notion of distance $\textrm{dist}(\mu, \nu)$ between probability measures. Using KL-type divergences lead to natural-gradient descent. One can use the Wasserstein distance, but this can be computationally difficult (although there is quite a bit of work to make this practical). The manuscript proposes using the sliced-Wasserstein distance: since the sliced-Wasserstein distance is relatively straightforward to approximate with Monte-Carlo samples, this leads to a practical method. The manuscript focusses on two common situations, $$\mathcal{F}(\mu) = \int V(x) \mu(dx) + \textrm{Entropy}(\mu)$$ and $$\mathcal{F}(\mu) = \iint W(x-y) \mu(dx) \otimes \mu(dy)$$ although the proposed methodology is quite general.

**Broader Impact Concerns:**

no concern.

**Requested Changes:**

# critical
I am not finding the method well motivated. Can the authors demonstrate, with appropriate simulations to back up their claims, that running a standard gradient-based optimizer (eg. ADAM) is much worse than what is proposed? The discussion could for example comment on the final accuracy, and the computational complexity.
* when comparing the different schemes, why do you make these choices for the number of time-steps $\tau$, the number of SW-JKO steps, etc.. For example, when comparing the time taken for different methods, how to you choose when to stop the different methods: wouldn't it make more sense to choose less examples and display the accuracy-versus-compute-time curves?
* Can the author compare to slightly more complicated but tractable situations? For example, mixture of Gaussians in not-so-small dimensions? Also, why choose ULA as competitor when there are many other better methods? Why choose these tuning parameters when using ULA?

# Minor:
* p4,"For a diffusion of length $k$, it requires $O(k^2)$ evaluations of gradients": what diffusion?
* most of page 6 are conjectures: it would make sense to make it much more explicit.
* " for randomly generated positive semi-definite matrices": what randomness? Would it be more interesting to study some ill-conditioned situations instead?
* in the Bayesian logistic regression example, what is the "accuracy" metric?
* in Section 4.2, I do not think it is true that the *centred* Dirac ring of radius 0.5 is the only solution: *any* Dirac ring of radius 0.5 is solution.
* is the example (ie. Aggregation Equation) in Section 4.2 a good example? Are the authors claiming that the proposed scheme converges faster to a solution? Or to a more accurate solution? (if so we would need some metric instead of nice pictures). As mentioned above, 10 lines of JAX and batch-gradient-descent give similar pictures orders of magnitude faster (and, it seems, as accurately).

# minor technical question:
The proof of the Proposition 1 and 2 are very simple since most of the hard parts are contained in [Candau-Tilh, 2020]. I am wondering whether one can apply the (reverse) triangular to the **squared** slice-Wasserstein distance $\text{SW}_2^2$. Should we apply this to the slice-Wasserstein distance $\text{SW}_2$ instead?

# Conclusion:
I think that the authors should:
* motivate much carefully their proposed method. Eg. does it lead to faster convergence when compared to simple baselines? more accurate solutions? etc..
* describe more carefully how the tuning parameters are chosen in the numerical experiments (and how the competing methods are tuned)
* choose (tractable) examples where proper metrics (ie. more reliable than FID score or pretty-looking pictures) are available. Also, as acknowledged by the authors, the results presented in Section 4.3 are not really competitive: it may be best to choose better examples where the method shines. As described in the introduction, there are **many** situations in applied mathematics that boil down to minimizing a functional over the space of probability distribution: can the authors choose a few well-chosen examples and convincingly show that the proposed method is useful: it is not the case at the moment.




**Strengths And Weaknesses:**

# Strength
I think that the text describes quite well and in an intuitive manner the current state-of-the-art regarding Wasserstein gradient flow. Furthermore, although I am not an expert in this area and do not know the literature well, the method indeed seems to not have been studied in the literature, with the exception of [LS2019]. The method and applications presented in [LS2019] are very different from what is presented in the manuscript.

# Weaknesses:

the manuscript describes three possible ways of parametrizing the probability distributions:

* **Grid:** for some fixed samples $x_1, \ldots, x_N$ consider $\mu = \sum \rho_i \delta_{x_i}$ for some positive weights $\rho_i$ that sum-up to one (and optimise these probability weights).
* **Particles:** consider $\mu = (1/N) \sum \delta_{x_i}$ for some free particle locations $x_1, \ldots, x_N$
* **Generative-models:** consider $\mu = g_{\theta} \sharp p_Z$ for some neural-net $g_{\theta}$ and some (Gaussian) base probability measure $p_Z$. If it is necessary to be able to evaluate the density of $\mu$ (for example to evaluate its entropy), one can use a normalizing flow.

With these parametrizations, the minimization of $\mu \mapsto \mathcal{F}(\mu)$ is quite straightforward. Something that I have not found clear at all in the text is the motivation for the introduction of the "regularisation" term $\frac{1}{2 \tau}\text{dist}^2(\mu, \mu_k)$. Indeed, after discretization, one can try to directly minimize $\mu \mapsto \mathcal{F}(\mu)$ with a standard gradient-based optimizer. Indeed, the usual argument is that the regularisation term $\frac{1}{2 \tau}\text{dist}^2(\mu, \mu_k)$ helps "take the geometry of the problem into account" and it is definitely true in some cases (eg. natural gradient, etc..) that this can greatly improve upon vanilla gradient descent. And I think I was expecting a small discussion, with some appropriate simulations. As a matter of fact, I have tried to reproduce some of the experiments in the paper: one can minimize with batch-gradient-descent the function $\mathcal{F}(\mu) = \iint W(x-y) \mu(dx) \otimes \mu(dy)$ described in the paper with the "particles"-parametrization proposed in the paper with 10 lines of JAX. This works very well and I could make things converge with $N=10^3$ particles in less than a second on a laptop (the manuscript reports running experiments for up to 5 hours). And I did obtain a result very similar to what is reported in the paper.


# References
[LS2019] Liutkus A, Simsekli U, Majewski S, Durmus A, Stöter FR. Sliced-Wasserstein flows: Nonparametric generative modeling via optimal transport and diffusions. InInternational Conference on Machine Learning 2019 May 24 (pp. 4104-4113). PMLR.

---

> ### Author Response · Authors · 2022-08-19
> **Response to Reviewer zKX6 (Part 1)**
>
> We thank the reviewer for their comments, and the positive reception of the introduction of the state of the art of Wasserstein gradient flows. Below, we address their questions and concerns.
>
> **The method indeed seems to not have been studied in the literature, with the exception of [1].** This method has not yet been studied in the litterature. Note that, as described *e.g. in Section 3*, [1] studies a different problem than us. To be more precise, they propose a SDE which is an approximation of the *Wasserstein* gradient flow of SW. While in our paper, we study the *sliced-Wasserstein* gradient flow of any functional. Hence, the two works are fundamentally different.
>
> **Something that I have not found clear at all in the text is the motivation for the introduction of the "regularisation" term $\frac{1}{2\tau}dist^2(\mu,\mu_k)$.** Maybe the motivation was not clearly presented in the paper. We can see this distance term as a regularization term, and it actually provides less numerical issues in practice (see e.g. Section 4.1., paragraph “Gaussian case”). However, adding this regularization from a theoretical point of view allows us to properly define gradient flows of probability measures, hence guaranteeing that we obtain the correct minimum of the functional, at least theoretically. This particular term allows indeed to define a particular type of proximal operator, defined on the set of probability measures, and we refer the reviewer to the large literature on proximal methods in Hilbert spaces to have a broader view of its interest in non-smooth optimization.  Notice that those schemes have been extensively studied for Wasserstein gradient flows, but since the SW space is not a geodesic space, the theoretical study of SW gradient flows is more complicated and we hope that the experimental results we provided will stimulate the community to study its theoretical properties. However, also notice that the scheme at each step is well defined for many functionals (see Proposition 1).
>
> **As a matter of fact, I have tried to reproduce some of the experiments in the paper: one can minimize with batch-gradient-descent the function $F(\mu)=\iint W(x-y)\mathrm{d}\mu(x)\mathrm{d}\mu(y)$ described in the paper with the "particles"-parametrization proposed in the paper with 10 lines of JAX. This works very well and I could make things converge with $N=10^3$ particles in less than a second on a laptop (the manuscript reports running experiments for up to 5 hours). And I did obtain a result very similar to what is reported in the paper.**
>
> Actually, the interest of this experiment was to show that we can obtain the right limit using the SW-JKO scheme with any neural networks while JKO-ICNN suffers from instabilities (see Figure 4) because of the use of ICNNs. Note that these instabilities were also reported in a concomitant work [2] (see Appendix D.3).
>
> It is possible that minimizing directly the functional over particles would give the right result. But note that from a theoretical point of view, it is not defined as a gradient flow over probability measures. To be more precise, it is a gradient descent in the space of parameters used to model the probability measures and not a gradient flow in the space of probabilities. However, we agree that since we need to model the probability measures in practice, we also need to do gradient descent in the space of parameters of the model to solve each step of the JKO scheme, which is a nonconvex problem.
>
> **Can the authors demonstrate, with appropriate simulations to back up their claims, that running a standard gradient-based optimizer (eg. ADAM) is much worse than what is proposed?**
>
> It is difficult to show that we always obtain better results compared to a direct optimization. Indeed, while JKO schemes allow to obtain well defined theoretical results, in practice, it is needed to model probability measures, and then to perform a gradient descent on weights, which is a nonconvex problem. Hence, the results might not always be better. But we still noticed that training normalizing flows with the SW-JKO scheme is less sensitive to numerical instabilities.
>
> Nevertheless, note that one key interest of the SW-JKO scheme is to provide a theoretical framework in which we could study the optimization over probability measures. But, since we need at some point to model these probability measures with neural networks (or particles or grid), and optimize over their weights (or the position of the particles), the problem becomes nonconvex and we suffer from the same problems of classical non convex optimization problems.
>
> We refer to the general response for more insights about the comparison between directly minimizing the functional and JKO schemes.

---

> > ### Author Response · Authors · 2022-08-19
> > **Response to Reviewer zKX6 (Part 2)**
> >
> > **When comparing the different schemes, why do you make these choices for the number of time-steps $\tau$, the number of SW-JKO steps, etc...**
> >
> > For Bayesian logistic regression, we followed the setup of [3] and made a grid search over parameters.
> >
> > For the Gaussian stationary solution, we fixed $\tau$ to have approximately the same dynamic between JKO-ICNN and SW-JKO with the same number of steps (i.e. we choose the same $\tau$ and use equation (18) for the SW-JKO scheme in order to take into account the dialtion, we refer to Appendix D.2). Hence, the comparison of the computational time and of the solution on these experiments makes sense.
> >
> > For the Aggregation equation, we refer to Appendix D.4. To sum up, we chose $\tau=0.5$ and 200 steps (to simulate a dynamic between $t=0$ and $t=10$). To take into the computational time of training JKO-ICNN, we choose $\tau=0.1$ and 100 steps for JKO-ICNN, hence simulating the dynamic of Wasserstein gradient flows between  $t=0$ and $t=10$.
> >
> >
> > **Can the author compare to slightly more complicated but tractable situations? For example, mixture of Gaussians in not-so-small dimensions?** We believe that using more complicated distributions such as a mixture of Gaussians would not bring much more since it will mostly depend on the expressiveness of the used normalizing flows (since the basic ones are known to have trouble recovering multi modal distributions, see e.g. [4,5]). We refer to the general response for a more thorough discussion.
> >
> > **Also, why choose ULA as competitor when there are many other better methods? Why choose these tuning parameters when using ULA?**
> >
> > We choose ULA as a competitor in Section 4.1 as ULA is an approximation of the Wasserstein gradient flows. The main competitor (when we first wrote this paper) was JKO-ICNN which we compare with for the bayesian logistic regression and the aggregation equation. For the stationary solution for Gaussians, we do not provide the results of JKO-ICNN since we encountered many numerical instabilities at some time (see Figure 2, right).
> >
> > The learning rate for ULA was chosen arbitrary (following concurrent works such as [3]) and the number of step was chosen to approximate the solution of the Fokker-Planck equation at the same time as the number of SW-JKO steps performed.
> >
> >
> > **p4,"For a diffusion of length k, it requires $O(k^2)$  evaluations of gradients": what diffusion?** We agree that this term may be confusing as it is not a SDE. We will correct it to “For a JKO scheme of $k$ steps”.
> >
> > **Most of page 6 are conjectures: it would make sense to make it much more explicit.** We believe that we already make it pretty explicit. For example, in the paragraph Limit PDE, it is stated *“he conjectures that there is a correlation between the two gradient flows. We identify here some cases for which we can relate the Sliced-Wasserstein gradient flows to the Wasserstein gradient flows.“*, and in Section 3.2, *“Nevertheless, we conjecture that in the limit $t\to\infty”, SWGFs converge toward the same measure as for WGFs. We will study it empirically in Section 4 by showing that we are able to find as good minima as WGFs for different functionals.”*
> >
> > **"for randomly generated positive semi-definite matrices": what randomness? Would it be more interesting to study some ill-conditioned situations instead?**
> >
> > To have a robust experiment, we generated 15 different covariance matrix from the scikit-learn package with the “make_spd_matrix” function. More precisely, they draw first a matrix $A$ with independent uniform entries on $[0,1]$, then take the eigenvector from $AA^T$ and return a matrix $ADA^T$ with $D=I+\Delta$, with $\Delta$ a diagonal matrix composed of uniform values on $[0,1]$. We refer to scikit-learn for the code.
> >
> > Studying ill-conditioned situations would definetly be interesting, but the main interest here was to compare with known methods which approximate Wasserstein gradient flows.

---

> > > ### Author Response · Authors · 2022-08-19
> > > **Response to Reviewer zKX6 (Part 3)**
> > >
> > > **In the Bayesian logistic regression example, what is the "accuracy" metric?** The accuracy metric is the percentage of well classified examples on the test set.
> > >
> > > **In Section 4.2, I do not think it is true that the centered Dirac ring of radius 0.5 is the only solution: any Dirac ring of radius 0.5 is solution.** We refer to [6] Section 4.3.1 which states *“We observe that the solution concentrates on a Dirac ring with radius 0.5 centered at the origin, recovering analytical results on the existence of a stable Dirac ring equilibrium for these values of a and b.“* We agree however that it is true only for the initial conditions used (here a centered Gaussian) and we will precise it in the paper.
> > >
> > > **Is the example (ie. Aggregation Equation) in Section 4.2 a good example? Are the authors claiming that the proposed scheme converges faster to a solution? Or to a more accurate solution? (if so we would need some metric instead of nice pictures).** The main purpose of this experiment is to show that one advantage of the SW-JKO scheme over JKO-ICNN is to be able to use any neural network, and hence being able to recover complicated shape such as a Dirac ring with a simple multilayer perceptron while ICNNs suffer from numerical instabilities for such target.
> > >
> > > **The proof of the Proposition 1 and 2 are very simple since most of the hard parts are contained in [Candau-Tilh, 2020]. I am wondering whether one can apply the (reverse) triangular to the squared slice-Wasserstein distance $SW_2^2$. Should we apply this to the slice-Wasserstein distance $SW_2$  instead?** Thank you for noticing this. Indeed, the reverse triangular inequality should be applied to $SW_2$ and not to $SW_2^2$. Then by continuity, we have the results. We will correct it in the revised version.
> > >
> > > **Motivate much carefully their proposed method. Eg. does it lead to faster convergence when compared to simple baselines? more accurate solutions? etc...** About the speed of the convergence, it comes back to compare the dynamic of the SW gradient flows with W gradient flow, which we discuss in Section 3.2. About the accuracy of the solution, we showed through our experiments that the SW-JKO scheme is more flexible than JKO-ICNN, and hence allows to use any neural networks, thus allowing to take into account inductive biases. The main example of this is in Section 4.2 with the aggregation equation where JKO-ICNN is not able to recover well the Dirac ring while a simple MLP with the SW-JKO scheme does.
> > >
> > > **The results presented in Section 4.3 are not really competitive: it may be best to choose better examples where the method shines.** Note that the experiment in Section 4.3 shows the benefits over related particle schemes which approximate the Wasserstein gradient flow. We argue that the results are not competitive with SOTA methods since the functional is SW and we use simple neural networks. Using more complicated functions such as the Jensen-Shannon divergence, would probably give better results, at the cost of using the variational formulation of [2] (which can be complementary to our work). We believe that it is more interesting to compare with related particle schemes proposed in [1], and first used on real data (MNIST, Fashion MNIST and CelebA), hence demonstrating the benefit of our scheme.
> > >
> > >
> > >
> > > [1] Liutkus, Antoine, et al. "Sliced-Wasserstein flows: Nonparametric generative modeling via optimal transport and diffusions." International Conference on Machine Learning. PMLR, 2019.
> > >
> > > [2] Fan, Jiaojiao, Amirhossein Taghvaei, and Yongxin Chen. "Variational Wasserstein gradient flow." arXiv preprint arXiv:2112.02424 (2021).
> > >
> > > [3] Mokrov, Petr, et al. "Large-scale wasserstein gradient flows." Advances in Neural Information Processing Systems 34 (2021): 15243-15256.
> > >
> > > [4] Dinh, Laurent, et al. "A RAD approach to deep mixture models." arXiv preprint arXiv:1903.07714 (2019).
> > >
> > > [5] Cornish, Rob, et al. "Relaxing bijectivity constraints with continuously indexed normalising flows." International conference on machine learning. PMLR, 2020.
> > >
> > > [6] Carrillo, José A., et al. "Primal dual methods for Wasserstein gradient flows." Foundations of Computational Mathematics 22.2 (2022): 389-443.

---

### Author Response · Authors · 2022-08-19
**General Response (Part 1)**

We thank the reviewers for their comments and their thorough reviews. We released a revised version of the paper with some modifications in red. Here, we will address common concerns among the reviewers.

The main concern of the reviewers is on the benefit of using any regularization versus directly minimizing the functional (i.e. $\mu\mapsto \mathcal{F}(\mu)$). While the latter can work in practice by modeling the probability measure and performing a gradient descent on the weights, it is not theoretically well defined as a gradient flow in the space of probability measures. In contrast, endowing the space with a distance between distributions allows to benefit from many theoretical guarantees. This has been studied extensively for Wasserstein gradient flows. Here, we proposed sliced-Wasserstein gradient flows, we showed empirically that they work as well as Wasserstein gradient flows and we underlined empirically some connections with classical Wasserstein gradient flows. For now, finding the characterization of the sliced-Wasserstein gradient flows is out of reach but we hope to raise interest on this very interesting theoretical question.

We believe that the main goals of the paper were not clearly stated and we rewrote the introduction of Section 2 in the revised version of the paper. To sum up, the goals are twofold, first, we aim at minimizing functions in the space endowed by SW by using a JKO like scheme (since SW is more computationally friendly). Second, we aim at studying the dynamics obtained with these new gradient flows and we draw connections with Wasserstein gradient flows. Hence, we believe that the comparison with directly minimizing the functional, though definitely relevant, is a little bit out of scope of the objectives of the paper.

However, it is an interesting question and we performed some comparisons with directly minimizing the functional. First, we compared training normalizing flows with and without the SW regularization (for the reverse KL/Fokker-Planck functional) and we found that adding the regularization is beneficial to avoid numerical instabilities during the training, especially when augmenting the dimension. On the quality of the solution, we believe that it is a complicated question to answer since, at the end of the day, we also require to model the probability measure in practice, and to perform a gradient descent over parameters, which will not be convex. Moreover, we argue that it is complicated to compare fairly both JKO type methods and a direct minimization method since the objectives are different, and hence, choosing comparable hyperparameters is not straightforward (the direct minimization will necessarily converge faster since it is the final objective, when JKO scheme might need many steps to reach the stationary solution).

We provide here some results for the bayesian logistic regression. We see that in some cases, we obtained better results with the regularization, hence avoiding local minima. However, on other examples, minimizing directly the functional also outperforms JKO-ICNN which raises questions on the benefits of these types of methods in practice.

|Dataset \ Method |JKO-ICNN|SWGF+Real NVP| Direct Minimization |
|-|--------|-------------|---------------------|
| covtype | **0.755** | **0.755** | 0.753 |
| german | 0.679 | **0.68** | 0.65 |
| diabetis | 0.777 | 0.778 | **0.786** |
| twonorm | **0.981** | **0.981** | 0.98 |
| ringorm | 0.736 | **0.741** | 0.738 |
| banana | 0.55 | 0.559 | **0.573** |
| splice | 0.847 | **0.85** | 0.848 |
| waveform | **0.782** | 0.776 | 0.77 |
| image | **0.822** | 0.821 | 0.821 |

---

> ### Author Response · Authors · 2022-08-19
> **General Response (Part 2)**
>
> It was also requested to test the SW-JKO scheme on more difficult stationary solutions (with the Fokker-Planck functional) such as mixture of Gaussians. While it would be a relevant experiment, we believe that it would not add much compared to the experiments we carried out (i.e. target as Gaussian and bayesian logistic regression) since it would amount to finding better normalizing flows (or using variants such as [1,2]) or tricks to learn the prior (such as [3]) in order to take into account the multi-modality of the data. Yet, it is not really the main point of this paper.
>
> Finally, we would like the reviewers to take into account that this work studies empirically the SW gradient flows and raises many interesting theoretical questions such as: can the dynamic of SW gradient flows be described with a PDE? And is this dynamic related to Wasserstein gradient flows? We provide here an empirical study and we hope to raise interest to the community to answer these questions.
>
>
> [1] Dinh, Laurent, et al. "A RAD approach to deep mixture models." arXiv preprint arXiv:1903.07714 (2019).
>
> [2] Cornish, Rob, et al. "Relaxing bijectivity constraints with continuously indexed normalising flows." International conference on machine learning. PMLR, 2020.
>
> [3] Stimper, V., Schölkopf, B., & Hernández-Lobato, J. M. (2022, May). Resampling Base Distributions of Normalizing Flows. In International Conference on Artificial Intelligence and Statistics (pp. 4915-4936). PMLR.

---

### Decision · Action_Editors · 2022-10-27

**Recommendation:** Accept with minor revision

**Comment:**

The paper has now been reviewed by three reviewers: two lean towards reject (zKX6 and nXjA) and one leans towards acceptance (xUbb). All the reviewers agree that the method is theoretically grounded and it appears experimentally to perform better than JKO-ICCN. However, when one is interested in finding a minimiser of a functional defined on the space of probability measures (as in the experiments of the paper), there exist alternatives which are much simpler as emphasized by the reviewers and the proposed method does not always outperform them (as demonstrated by the experiments of the authors in the rebuttal). Those alternatives are not gradient flows in the space of probability measures but for most ML applications this is somewhat irrelevant.

I believe the material is worth publishing as it is a nice addition to the computational gradient flows literature and is well written. However the authors should follow closely the recommendations of zKX6 and update the "contribution" section of the paper by
- only stating that the proposed method is somehow more competitive that JKO-ICNN
- add some discussion regarding the competitiveness of the method when compared to the vanilla not-regularised approach.
- if something is about the dynamics of the flow, illustrate it more convincingly.

**Audience:**

There is a growing literature on gradient flows for optimizing functional of probability measures so some individuals in TMLR's audience will be interested about this novel method.

**Claims And Evidence:**

The authors propose a novel gradient flow scheme to minimize functional of probability measures. A standard approach is to use a JKO scheme, various groups have recently proposed to approximating numerically this scheme by using ICNNs. However, this is computationally costly. The authors propose here to replace the Wasserstein metric appearing in JKO by the sliced-Wasserstein metric which is computationally cheaper to evaluate and bypasses the use of ICNNs. This is an interesting and "natural" idea. The proposed method is principled and the claims of the authors about the computational complexity are substantiated.

---

> ### Author Response · Authors · 2022-11-03
> **Revision**
>
> Dear Action Editors and Reviewers,
>
> Thank you for your comments.
>
> We submitted a new revision of the paper.
>
> Following the comments, we modified the “Contributions” section of the paper by stating that our method performs better than JKO-ICNN. We also added at the end of Section 3 a discussion about the minimization approach without regularization.
>
> Concerning the dynamics of the flows, additionally to Figure 1 in which we compare trajectories of the Wasserstein and Sliced-Wasserstein (SW) gradient flows of particles, we have in Appendix D.1 comparisons of the evolution of a functional along its true Wasserstein gradient flow and the learned SW gradient flow. We believe that this discussion better fits in the Appendices of the paper, but if you and the reviewers believe that it should belong to the main body of the paper, we can move them to Section 3.2
>
> Best regards